# PrefixQuant: Static Quantization Beats Dynamic through Prefixed Outliers in LLMs

## Abstract

Quantization is essential for deploying Large Language Models (LLMs) by enhancing memory efficiency and inference speed. Existing methods for activation quantization mainly address channel-wise outliers, often neglecting token-wise outliers, leading to reliance on costly per-token dynamic quantization. To address this, we introduce PrefixQuant, a novel technique that isolates outlier tokens offline without re-training. Specifically, PrefixQuant identifies high-frequency outlier tokens and prefixes them in the KV cache, preventing the generation of outlier tokens during inference and simplifying quantization. To our knowledge, PrefixQuant is the first to enable efficient per-tensor static quantization to outperform expensive per-token dynamic quantization. For instance, in W4A4KV4 (4-bit weight, 4-bit activation, and 4-bit KV cache) Llama-3-8B, PrefixQuant with per-tensor static quantization achieves a 7.43 WikiText2 perplexity and 71.08% average accuracy on 5 common-sense reasoning tasks, outperforming previous per-token dynamic quantization methods like QuaRot with 0.98 perplexity improvement and +5.98 points accuracy. Additionally, the inference speed of W4A4 quantized models using PrefixQuant is $1.60\times$ to $2.81\times$ faster than FP16 models and exceeds QuaRot models by $1.2\times$ to $1.3\times$.

## 1 Introduction

Recently, Large Language Models (LLMs)(Touvron et al., 2023; Bubeck et al., 2023) demonstrate remarkable capabilities across various tasks, significantly improving the convenience of daily work and life. However, their large parameters and computational demands pose significant challenges for deployment. This makes quantization (Frantar et al., 2022; Lin et al., 2023; Shao et al., 2023) a crucial technology for reducing memory usage and speeding up inference (Yuan et al., 2024).

Despite advancements, large outliers in LLMs activations can lead to significant quantization errors and accuracy loss. Many current methods address this by focusing on alleviating channel-wise outliers (Dettmers et al., 2022) through techniques like channel-wise scaling (Xiao et al., 2023a; Shao et al., 2023; Wei et al., 2023a), mixed-precision quantization (Dettmers et al., 2022; Zhao et al., 2023), Hadamard rotation (Ashkboos et al., 2024b; Liu et al., 2024a), and channel-level assembly (Liu et al., 2023). However, activations of LLMs include not only channel-wise but also token-wise outliers. For example, Figure 1 (a) shows that some tokens, can be termed as outlier tokens, have extreme values exceeding 1,000, making it impractical to share quantization scales between outlier and normal tokens. The current leading method, Hadamard rotation (Ashkboos et al., 2024b), redistributes outlier values across all channels, reducing the maximum value in outlier tokens from over 1,000 to about 15 (see Figure 1 (b)). Nevertheless, the magnitude of outlier tokens remains hundreds of times greater than that of normal tokens, still suffering significant performance degradation when sharing quantization scales across different tokens.

Due to such dramatic discrepancies between normal and outlier tokens, previous quantization methods have to rely on per-token dynamic quantization to adjust quantization scales on-the-fly for each token. While per-token dynamic quantization adapts better to distribution changes, it faces more computational effort (Xiao et al., 2023a) and less compatible with operator fusion (Nagel et al., 2021) than per-tensor static quantization which use a fixed quantization parameter for all token. This leads to an important question: ***Can we eliminate token-wise outliers to enhance the precision of efficient per-tensor static quantization?***

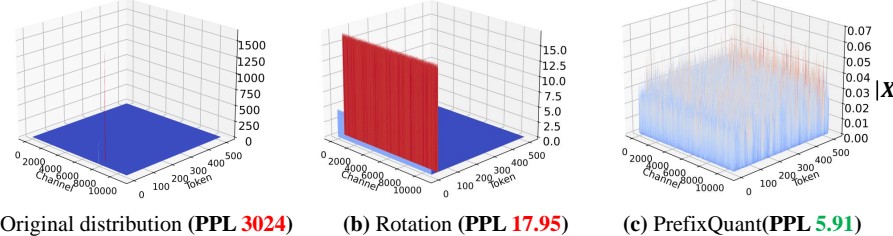

**(a)** Original distribution (**PPL 3024**)     **(b)** Rotation (**PPL 17.95**)     **(c)** PrefixQuant(**PPL 5.91**)

Figure 1: **Comparison of PrefixQuant with existing methods.** This figure shows the input activation of the down_proj linear layer in Llama-2-7B using different methods. Perplexity is measured with Llama-2-7B under 16-bit weight and 4-bit activation using per-tensor static quantization without any re-training. The original distribution has significant outliers larger than 1,000 (left). The previous method with Hadamard rotation (Ashkboos et al., 2024b) reduces outliers to nearly 15 (middle) but still suffers from poor perplexity due to a non-uniform distribution. We propose PrefixQuant (right), which prefixes some specific tokens in KV cache to isolate outliers, reducing the maximum to nearly 0.07, significantly improving quantization performance.

In this paper, we propose PrefixQuant, an efficient solution for static activation quantization in LLMs. PrefixQuant is based on a key observation: outlier tokens usually appear at fixed positions in the token sequence (such as the initial token) or in tokens with low semantic value (such as "\n", ".", "the", *etc*). Based on this observation, PrefixQuant pre-processes the outlier tokens offline in the KV cache to prevent generate new outlier tokens during inference. Specifically, given a LLM, PrefixQuant firstly counts the number $N$ of outlier tokens, and selects the Top-$N$ high-frequency outlier tokens to prefix in the KV cache. This process is efficient and does not require any retraining, unlike previous methods (Sun et al., 2024; Bondarenko et al., 2024), and can be completed quickly, such as in 12 seconds for Llama-2-7B. As illustrated in Figure 1 (c), PrefixQuant effectively eliminates outlier tokens, achieving excellent performance with per-tensor static activation quantization. For example, with 4-bit per-tensor static activation quantization on Llama-2-7B, PrefixQuant achieves 5.91 perplexity, significantly outperforms QuaRot which has a perplexity of 17.95. Furthermore, we introduce a block-wise fine-tuning optimization (Shao et al., 2023; Chen et al., 2024a) to improve performance by simultaneously training the quantization parameters of both weight and activation. Additionally, we also find that isolating the outlier tokens enhances the convergence stability of training through avoiding large outliers magnitude during the calculation of Mean Square Error (MSE) loss. Thus, the proposed method of prefixed outliers can also serve as a plug-and-play enhancement for existing optimization-based quantization methods (Shao et al., 2023; Chen et al., 2024a).

Experiments demonstrate that, without any fine-tuning, PrefixQuant achieves comparable or better performance than previous per-token dynamic quantization methods (Ashkboos et al., 2024b; Xiao et al., 2023a; Lin et al., 2024b) using coarser per-tensor static quantization. Furthermore, fine-tuning significantly enhances PrefixQuant's performance. For example, PrefixQuant with fine-tuning achieves a 7.43 WikiText2 (Merity et al., 2016) perplexity and 71.08% average accuracy across five common-sense reasoning tasks in W4A4KV4 Llama-3-8B, significantly outperforming previous QuaRot (Ashkboos et al., 2024b) with 0.98 perplexity benefit and +5.98 points accuracy. To the best of our knowledge, PrefixQuant is the first to outperform previous per-token dynamic quantization methods (Ashkboos et al., 2024b; Xiao et al., 2023a; Lin et al., 2024b) using coarse per-tensor static quantization. We also benchmark the end-to-end inference of W4A4 quantization, where PrefixQuant achieves a $1.60\times$ to $2.81\times$ speedup over FP16 models, and surpasses QuaRot models by $1.2\times$ to $1.3\times$. We hope PrefixQuant inspires future developments in LLM compression.

## 2 RELATED WORKS

In this section, we discuss works related to outliers in LLMs, including quantization methods that enhance performance by eliminating activation outliers. We divide the discussion into channel-wise and token-wise outliers.

**Channel-Wise Outliers.** Dettmers et al. (2022) identifies that outliers in activation consistently occur in the same channels across different input tokens and proposes isolating these outlier channels

with 16-bit precision. Other works, such as Atom (Zhao et al., 2023) and QUIK (Ashkboos et al., 2023), follow a similar mixed-precision approach to handle outliers. Instead of introducing mixed-precision matrix manipulation, which lacks native hardware support, another line of work addresses outliers through mathematically equivalent transformations. For example, SmoothQuant (Xiao et al., 2023a), OmniQuant (Shao et al., 2023), and Outlier Suppression (Wei et al., 2022; 2023b) mitigate outliers by scaling activations to weights on a channel-wise basis. QLLM (Liu et al., 2023) reduces outlier values by dividing each outlier channel into multiple sub-channels. Recently, QuaRot (Ashk-boos et al., 2024b) proposed a simple and effective method, random Hadamard rotation, to redis-tribute outliers across all channels. Building on QuaRot, SpinQuant (Liu et al., 2024a) suggests training the orthogonal matrix instead of using a random Hadamard matrix to further enhance per-formance. DuQuant (Lin et al., 2024a) leverages channel permutation to evenly distribute outlier to each block and uses block-rotation to smoothen outliers. Although these methods significantly improve activation quantization performance, they all rely on fine-grained per-token dynamic quan-tization, which incurs additional overhead to manage token-wise fluctuations.

**Token-Wise Outliers.** The SoftMax function used in the self-attention mechanism naturally pre-vents producing zero attention scores. As a result, the model tends to assign unnecessary scores to special tokens, leading to token-wise outliers (or termed as massive activation) (Sun et al., 2024; Xiao et al., 2023b). Based on this, StreamingLLM (Xiao et al., 2023b) and LM-infinite (Han et al., 2023) support infinite sequences by retaining the initial token. Unlike StreamingLLM and LM-infinite, which simply preserve initial tokens in the KV-cache for long-context generation, our Pre-fixQuant carefully selects prefixed tokens in the KV-cache to isolate outliers for quantization. Some studies explore eliminating outliers with training techniques. For example, Bondarenko et al. (2024) allows SoftMax to produce zero values, and Sun et al. (2024) shows that adding attention bias in the KV cache during training can effectively reduce outliers. Our PrefixQuant efficiently isolates outlier tokens without needing retraining. The works closest to our approach are QFeP (Yang et al., 2024) and CushionCache (Son et al., 2024), which also set prefixed tokens in the KV cache. However, CushionCache (Son et al., 2024) takes 12 hours to find the prefixed tokens for Llama-3-8B through a greedy search, while our method completes this process in 12 seconds. QFeP (Yang et al., 2024) fixes the outlier token number for all models at 3, which lacks flexibility. Additionally, both QFeP and CushionCache suffer significant performance degradation when using per-tensor static quanti-zation instead of per-token dynamic quantization. Our PrefixQuant is the first to make per-tensor static quantization outperform per-token dynamic quantization.

## 3 PRELIMINARIES

Quantization in LLMs involves weight, activation, and KV cache quantization. Weight quantiza-tion (Chen et al., 2024a) and KV cache quantization (Liu et al., 2024b) reduce memory usage and speed up memory-bound computations (Yuan et al., 2024). Combining weight and activation quan-tization enables low-bit matrix manipulation to accelerate computation-bound tasks (Yuan et al., 2024). Specifically, the symmetric quantization process is:

$$\mathbf{X}_{\text{INT}} = \text{clamp}(\lfloor \frac{\mathbf{X}}{\mathbf{s}_{\text{X}}} \rceil, -2^{N-1}, 2^{N-1} - 1), \tag{1}$$

where $\lfloor \cdot \rceil$ denotes rounding operation, $N$ is the target bit number, $\mathbf{X}_{\text{INT}}$ and $\mathbf{X}$ are the quantized integer and full-precision activation, respectively. $\mathbf{s}_{\text{X}}$ is the step size. Full precision weight $\mathbf{W}$ can also be quantized into $\mathbf{W}_{\text{INT}}$ and $\mathbf{s}_{\text{W}}$ similarly. Then, full-precision matrix manipulation transfer into efficient low-bit matrix manipulation:

$$\mathbf{X}\mathbf{W} \approx (\mathbf{s}_{\text{W}} \cdot \mathbf{s}_{\text{X}}) \cdot \mathbf{X}_{\text{INT}}\mathbf{W}_{\text{INT}} \tag{2}$$

**Granularity.** Finer granularity in quantization results in more overhead but less information loss. Per-tensor quantization shares $\mathbf{s}$ across the entire tensor. Per-channel quantization of weight and per-token quantization of activation means $\mathbf{s}$ is shared within each row of the tensor.

**Dynamic and Static.** Activation quantization divides into dynamic and static quantization based on how quantization parameters are calculated. Specifically, dynamic quantization calculates $\mathbf{s}_{\text{X}} = \frac{\max(|X|)}{2^{N-1}-1}$ during inference, offering better adaptability to different distributions. In contrast, static quantization precomputes $\mathbf{s}_{\text{X}}$ and $(\mathbf{s}_{\text{W}} \cdot \mathbf{s}_{\text{X}})$ in Eq.(2) offline, leading to more efficient inference

and more feasible operator fusion (Nagel et al., 2021). Table 8 shows that the overhead of static quantization is nearly $3\times$ lower than dynamic quantization. Additionally, we initialize both $\mathbf{s}_W$ and $\mathbf{s}_X$ through grid search (Lin et al., 2023; Gong et al., 2024) on a small calibration dataset for all experiments with static quantization.

**Hadamard Rotation.** Random Hadamard rotation (Ashkboos et al., 2024b; Liu et al., 2024a) addresses channel-wise outliers. Our method focus on removing token-wise outliers. Therefore, We build our method upon the Hadamard rotation technique, and the detailed is provided in Sec. C.

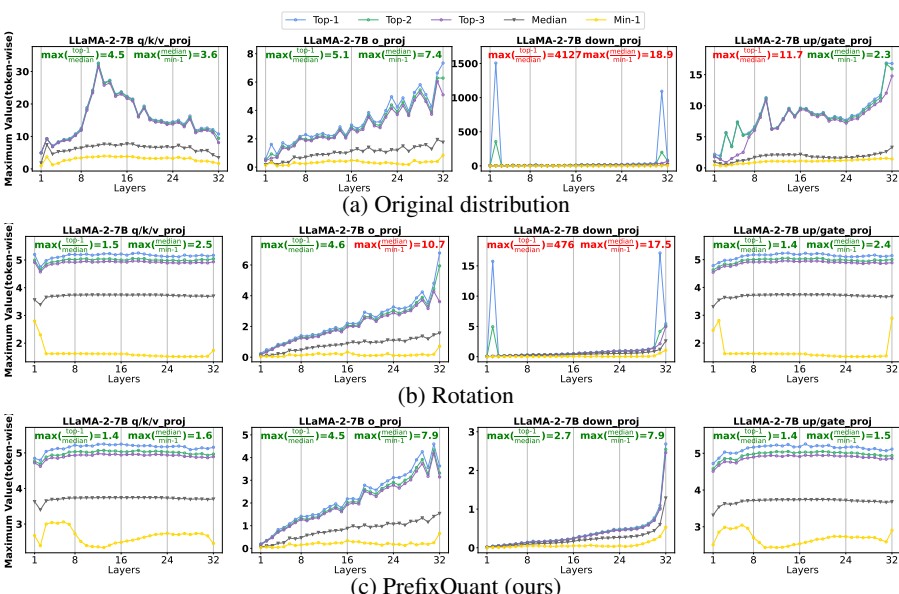

(a) Original distribution

(b) Rotation

(c) PrefixQuant (ours)

Figure 2: **Distribution of token-wise maximum values for linear layers inputs in Llama-2-7B.** Top-$N$ indicates the $N$-th largest value, Min-$N$ indicates the $N$-th smallest value. We also report the maximum ratio between Top-1 value and median value, as well as the maximum ratio between median value and Min-1 value (Ratios greater than 10 are marked with red, and the rest are green). Lower ratio indicate similar maximum values across different tokens, leading compatibility with per-tensor static activation quantization.

## 4 DIFFICULTY OF STATIC QUANTIZATION

Both channel-wise and token-wise outliers can cause information loss during quantization. While channel-wise outliers have been thoroughly explored and addressed in prior research (Ashkboos et al., 2024b), this discussion focuses on token-wise outliers, which occur within specific tokens.

Let $\mathbf{X} \in \mathbb{R}^{T \times C}$ represent the token sequence, with $T$ tokens and a dimension size of $C$. We calculate token-wise maximum values $\mathbf{M} \in \mathbb{R}^T$, indicating the maximum value of each token. Per-tensor static quantization uses one pre-computed scale for all tokens. If the token-wise maximum values $\mathbf{M}$ vary significantly across tokens, this can lead to substantial information loss after per-tensor static quantization. To analyze the distribution of token-wise maximum values $\mathbf{M}$ and understand the challenges for per-tensor static quantization, we define top-1, median, and min-1 as the largest, median, and smallest values of $\mathbf{M}$, respectively. We then measure discrepancies using the ratios $\frac{\text{top-1}}{\text{median}}$ and $\frac{\text{median}}{\text{min-1}}$. Specifically, a larger $\frac{\text{top-1}}{\text{median}}$ indicates upper outliers, while a larger $\frac{\text{median}}{\text{min-1}}$ represents lower outliers. Both ratios highlight the variability in $\mathbf{M}$. Specifically, we identify the following patterns that motivate our method.

**1. Upper Outlier Tokens in Inputs.** As shown in Figure 2a, the input activation of down_proj layers exhibits significant outliers with $\frac{\text{top-1}}{\text{median}} = 4127$. Although Hadamard rotation (Figure 2b) reduces the ratio to 478, it remains impractical to share a quantization scaling factor across tokens due to the large gap in maximum values.

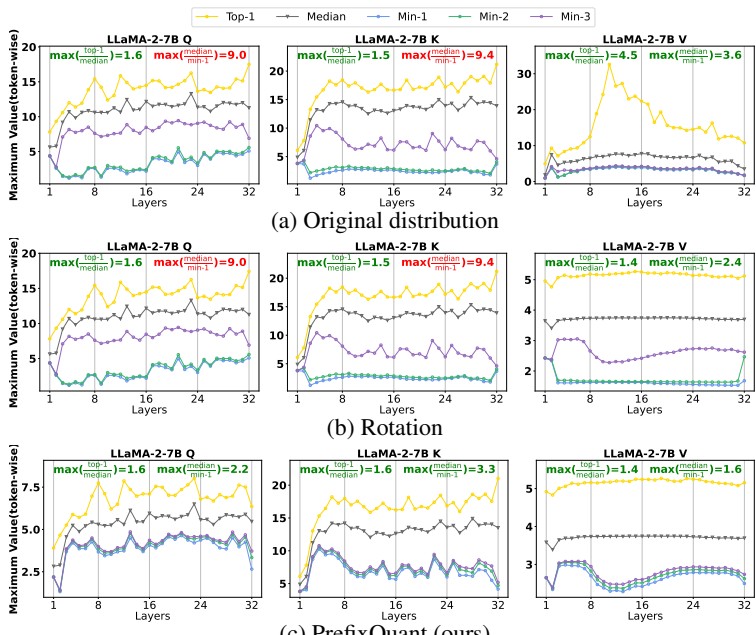

(a) Original distribution

(b) Rotation

(c) PrefixQuant (ours)

Figure 3: **Distribution of token-wise maximum values for Q/K/V in Llama-2-7B.** Same present rules as Figure 2a except that ratios greater than 5 are marked with red.

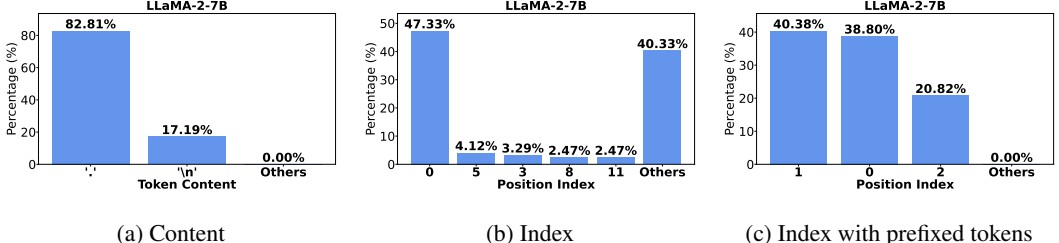

(a) Content        (b) Index        (c) Index with prefixed tokens

Figure 4: **Illustration the content and index of outlier token in the input sequence of Llama-2-7B.** (a) counts the outlier tokens except in the initial token, shows that the outliers only exit in "." and "\n" tokens. (b) illustrates the sequence index of outlier tokens. (c) demonstrates that prefix the input sequence with ".\n[BOS]" can constrain the outlier token in the first three tokens.

**2. Lower Outlier Tokens in Q/K/V.** We also investigate the distribution of **Q/K/V** within the self-attention mechanism. We only quantize the KV cache for fair comparisons with previous works (Ashkboos et al., 2024b; Lin et al., 2024b). However, quantization of **Q** is also crucial, as used in FlashAttention-3 (Shah et al., 2024). In Figure 3, **Q/K/V** display a different outlier pattern than the inputs of linear layers, with some tokens having extremely small magnitudes instead of large ones. Specifically, **Q/K** have $\frac{\text{top-1}}{\text{median}} \approx 1.5$, but $\frac{\text{median}}{\text{min-1}} > 9$. Additionally, as shown in Figure 3b, Hadamard rotation has no effect on these outliers.

**3. Outlier Tokens in Initial or Low-Semantic Tokens.** Though outlier tokens occur in different patterns, we find that they are the same tokens in inputs of linear layers and **Q/K/V**. Consistent with Massive Attention (Sun et al., 2024), we find that outlier tokens appear only in small fractions (nearly 1 to 4 tokens in the input sequence) with fixed patterns. For example, Llama-2-7B has outlier tokens in both initial and delimiter tokens ("." or "\n" as shown in Figure 4a). However, unlike outlier channels that exist in some fixed channel indexes (Dettmers et al., 2022), the position indexes of outlier tokens relate to the input sequence and are diverse, as shown in Figure 4b. Therefore, it is not feasible to decide offline on the outlier token to achieve mixed-precision quantization like previous works on outlier channels (Dettmers et al., 2022; Zhao et al., 2023).

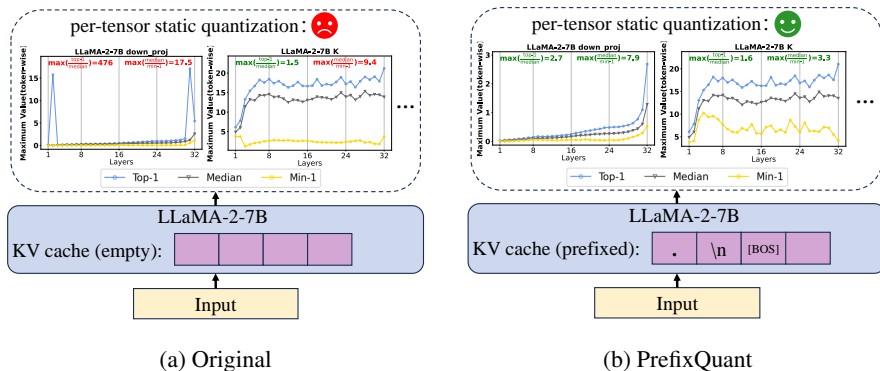

(a) Original          (b) PrefixQuant

Figure 5: **Comparison of Original and PrefixQuant Inference.** Both methods use Hadamard rotation to remove channel-wise outliers. PrefixQuant differs by setting specific prefixed tokens in the KV cache, which eliminates token-wise outliers in linear inputs and **Q/K/V**, enhancing compatibility with per-tensor static quantization. Llama-2-7B serves as an example; additional prefixed tokens for other models are listed in Table 1.

Previous works (Ashkboos et al., 2024b; Lin et al., 2024b; Liu et al., 2024b) take per-token dynamic quantization for inputs of linear layers and KV cache to deal with outlier tokens. In this paper, we focus on eliminating outlier tokens to facilitate per-tensor static quantization.

## 5   PREFIXQUANT

As shown in Figure 5, we propose prefixing outlier tokens in the KV cache to improve the performance of more efficient per-tensor static quantization, instead of using costly per-token dynamic quantization. Section 5.1 explains how to find these prefixed outliers. Section 5.2 introduces block-wise fine-tuning to further enhance performance.

### 5.1   PREFIXED OUTLIERS

**Definition of Outlier Token.** Given that both upper outlier tokens in the inputs of the linear layer and lower outlier tokens in **Q/K/V** are same tokens, we choose to identify outlier tokens using the upper outliers in the inputs of the down_proj layers due to the outlier in down_proj is more highlight and easier to be detected. Given token-wise maximum values $\mathbf{M} \in \mathbb{R}^T$, which represents the maximum values of each token. Then, outlier token in the $i$-th index of token sequence is identified when the ratio of their maximum values to the median of all maximum values exceeds a threshold $\eta$:

Table 1: Prefixed tokens in KV cache across different models. [BOS] indicates the special token for beginning of sequence(*e.g.* "" for Llama-2 and "|begin_of_text|" for Llama-3). Note that the following "_" represents space.

| Model | Prefixed token | |
|---|---|---|
| | **Number** | **Content** |
| Llama-2-7B | 3 | .\n[BOS] |
| Llama-2-13B | 3 | the.[BOS] |
| Llama-2-70B | 4 | \n"[BOS] |
| Llama-3-8B(-Instruct) | 1 | [BOS] |
| Llama-3-70B(-Instruct) | 3 | ,_[BOS] |
| Mistral-v0.3-7B | 4 | \n.to[BOS] |
| Qwen-2-7B | 1 | [BOS] |

$$\frac{\mathbf{M}_i}{\text{median}(\mathbf{M})} > \eta, \qquad (3)$$

where $\mathbf{M}_i$ is the maximum value of the $i$-th token, median() denotes the function to find the median value from the vector, and the threshold $\eta$ is empirically set to $64$ in our experiments.

**Number of Outlier Tokens.** We determine the number of outlier tokens by calculating the average number of outlier tokens in a small calibration dataset. Specifically, we compute the average outlier token count $\mathbf{O} \in \mathbb{R}^b$ for each transformer block according to Eq (3), where $b$ is the total number of transformer blocks. Since outlier tokens are nearly consistent across layers that contain them, we set the number of outlier tokens as $o = \lceil \max(\mathbf{O}) \rceil$.

| PPL ↓ | FP16 | W16A4KV16 (static) | | | W16A16KV4 (static) | | |
|---|---|---|---|---|---|---|---|
| | | original | + rotation | + prefixed | original | + rotation | + prefixed |
| Llama-2-7B | 5.47 | 3024.77 | 17.95 | **5.91** | 6.46 | 5.95 | **5.56** |
| Llama-3-8B | 6.14 | 1816.57 | 22.14 | **7.23** | 7.37 | 8.12 | **6.30** |

Table 2: Proposed prefixed outliers in KV cache significantly improves the performance of the static quantized models over hadamard rotation Ashkboos et al. (2024b); Liu et al. (2024a). W16A4KV16 indicates 4-bit per-tensor static quantization of all linear layer inputs. W16A16KV4 indicates 4-bit per-head static KV cache quantization. WikiText2 perplexity with 2048 context length is reported.

**Which Tokens to Prefix?** Outlier tokens act as attention sinks (Xiao et al., 2023b), occupying only a few tokens ($1 \sim 4$) to help the attention mechanism do nothing (Bondarenko et al., 2024; Sun et al., 2024). Given the outlier token number $o$, we find that prefixing the top-$o$ high-frequency[1] outlier tokens and the special '[BOS]' token can successfully constrains the outliers in prefixed tokens as shown in Figure 4c. For special models (such as Llama-3-8B and Qwen-2-7B) with outlier tokens only in the initial tokens, we simply set the prefix token as "[BOS]". The detailed prefixed tokens for different models are illustrate in Table 1. Considering the auto-regressive inference pipeline of LLMs, we store these prefix tokens in the KV cache to prevent generating new outlier tokens during inference. As shown in Figure 2c and Figure 3c, prefixing outliers in the KV cache reduces the $\frac{\text{top-1}}{\text{median}}$ ratio of down_proj inputs from 476 to 2.7 and the $\frac{\text{median}}{\text{min-1}}$ ratio of **Q/K** from $> 9$ to $< 3.5$.

**Quantitative Analysis.** Table 2 presents separate performance of static quantization on input activation and KV cache quantization. We can find that the model suffers significant performance degradation with static quantization becuase of outlier tokens. For example, in Llama-3-8B, WikiText2 perplexity increases from 6.14 to 22.14 with 4-bit per-tensor activation quantization and from 6.14 to 8.12 with 4-bit per-head static KV cache quantization even with Hadamard rotation. After further setting prefix outliers in the KV cache, performance significantly improves: perplexity of 4-bit per-tensor activation decreases to 7.23 and perplexity of 4-bit per-head static KV cache quantization decreases to 6.30, demonstrating the effectiveness of prefixed outlier tokens for static quantization.

## 5.2 BLOCK-WISE FINE-TUNING

Recent studies demonstrate that block-wise fine-tuning (Shao et al., 2023; Chen et al., 2024a) enhances performance by considering inter-layer interactions (Li et al., 2021). We initialize all quantization parameters using grid search (Lin et al., 2023; 2024b) and then fine-tune each transformer block with mean square error loss sequentially. For trainable parameters, we follow EfficientQAT (Chen et al., 2024a) by activating the training of all quantization parameters and original full-precision weights. Additionally, unlike dynamic activation quantization, the offline pre-computed quantization parameters of static activation quantization are inherently trainable. To maintain simplicity, we use block-wise quantization in this work and leave the end-to-end finr-tuning of EfficientQAT (Chen et al., 2024a) for future performance improvements.

# 6 EXPERIMENTS

## 6.1 SETUPS

**Baseline.** PrefixQuant is a versatile method applicable to any precision. We conduct experiments on three precisions: W8A8KV8, W4A8KV4, and W4A4KV4. In PrefixQuant, weight uses per-channel symmetric quantization. KV cache uses per-head symmetric static quantization for 4-bit and per-tensor symmetric static quantization for 8-bit. Activation (inputs of linear layers) uses per-tensor static quantization. We compare PrefixQuant with QuaRot (Ashkboos et al., 2024b), Atom (Zhao et al., 2023), DuQuant (Lin et al., 2024a), QoQ (Lin et al., 2024b), SmoothQuant (Xiao et al.,

---

[1]The frequencies are calculated without considering initial token.

Table 3: W4A4KV4 results. Perplexity is measured with context length 2048. "Avg. Acc." indicates the average zero-shot accuracy on 5 common-sense reasoning tasks. "Quant Type" is used to indicate whether the activation and kv cache quantization are dynamic or static.

| Model | Method | Quant Type | Wiki PPl | Avg. Acc. |
|---|---|---|---|---|
| Llama-2-7B | FP16 | - | 5.47 | 69.04 |
| | Atom | dynamic | 6.12 | 59.73 |
| | QuaRot | dynamic | 6.19 | 64.69 |
| | DuQuant | dynamic | 6.20 | 66.25 |
| | SpinQuant | dynamic | 5.95 | 65.35 |
| | PrefixQuant w/o FT | static | 6.22 | **66.84** |
| | PrefixQuant | static | **6.01** | 66.37 |
| Llama-2-13B | FP16 | - | 4.88 | 71.73 |
| | Atom | dynamic | 5.31 | 63.51 |
| | QuaRot | dynamic | 5.45 | 69.01 |
| | DuQuant | dynamic | 5.39 | 69.13 |
| | SpinQuant | dynamic | 5.24 | 69.24 |
| | PrefixQuant w/o FT | static | 5.50 | 69.92 |
| | PrefixQuant | static | **5.32** | **70.36** |
| Llama-2-70B | FP16 | - | 3.32 | 76.72 |
| | Atom | dynamic | **3.73** | 67.52 |
| | QuaRot | dynamic | 3.83 | 75.43 |
| | DuQuant | dynamic | 3.77 | 74.75 |
| | SpinQuant | dynamic | 3.70 | 75.19 |
| | PrefixQuant w/o FT | static | 4.41 | 73.29 |
| | PrefixQuant | static | 3.81 | **75.48** |
| Llama-3-8B | FP16 | - | 6.14 | 72.71 |
| | Atom | dynamic | 7.76 | - |
| | QuaRot | dynamic | 8.41 | 65.15 |
| | DuQuant | dynamic | 8.14 | 67.13 |
| | SpinQuant | dynamic | 7.36 | 68.23 |
| | PrefixQuant w/o FT | static | 7.93 | 68.37 |
| | PrefixQuant | static | **7.43** | **71.08** |
| Llama-3-70B | FP16 | - | 2.85 | 80.03 |
| | QuaRot | dynamic | 6.82 | 68.39 |
| | DuQuant | dynamic | 5.67 | 74.89 |
| | PrefixQuant w/o FT | static | 5.23 | 76.40 |
| | PrefixQuant | static | **4.41** | **77.18** |

\* Grayed results use Wikitext2 as calibaration dataset.
† Atom apply 128 group size quantization to both weight and activations.

2023a) and SpinQuant (Liu et al., 2024a). Following QoQ, we reproduce all these methods except SpinQuant with Pile (Gao et al., 2020) calibration dataset to avoid over-fitting for fair comparisons. The detailed quantization configuration and results sources of these comparison methods can be found at Sec. B. Note that all comparison methods use dynamic quantization if without specific mentioned, and would suffer dramatic performance degeneration likes "+ static quantization" in Table 6.

**Models and datasets.** We evaluate PrefixQuant on the Llama-2, Llama-3, Llama-3-Instruct families, Mistral-7B-v0.3, and Qwen-2-7B models. Following previous literature (Shao et al., 2023; Lin et al., 2024b), we assess PrefixQuant quantized models on language modeling and zero-shot tasks. Specifically, we evaluate on WikiText2 (Merity et al., 2016) with a 2048 context length for perplexity, and on PIQA (Bisk et al., 2020), ARC (Clark et al., 2018), HellaSwag (Zellers et al., 2019), and WinoGrande (Sakaguchi et al., 2021) using `lm_eval v0.4.2` (Gao et al., 2024). For accuracy, we report `acc` for WinoGrande and `acc_norm` for HellaSwag, Arc_Challenge, Arc_Easy, and PIQA, following Qserve (Lin et al., 2024b)[2].

**Grid Search Setting.** For all experiments with static quantization, we initialize the quantization parameters through grid search on 8 Pile (Gao et al., 2020) samples with a 1024 sequence length. We minimize the layer outputs for fine-grained quantization (per-channel/per-head) and block outputs

---

[2]Some weight-only quantization works such as EfficientQAT (Chen et al., 2024a) and QuiP# (Tseng et al., 2024) report `acc` for all tasks.

Table 4: W4A8KV4 results. Refer Table 3 for the metric setting and performance of full-precision models.

| Model | Method | Activation Quant | Wiki PPl | Avg. Acc. |
|---|---|---|---|---|
| Llama-2-7B | QoQ | dynamic | 5.75 | 67.22 |
| | QuaRot | dynamic | 5.73 | 67.11 |
| | PrefixQuant w/o FT | static | 5.76 | 67.86 |
| | PrefixQuant | static | **5.68** | **68.90** |
| Llama-2-13B | QoQ | dynamic | 5.12 | 70.56 |
| | QuaRot | dynamic | 5.07 | 69.96 |
| | PrefixQuant w/o FT | static | 5.08 | 71.07 |
| | PrefixQuant | static | **5.07** | **71.25** |
| Llama-2-70B | QoQ | dynamic | 3.52 | 75.91 |
| | QuaRot | dynamic | 3.46 | 76.31 |
| | PrefixQuant w/o FT | static | 3.60 | 75.00 |
| | PrefixQuant | static | 3.50 | 76.50 |
| Llama-3-8B | QoQ | dynamic | 6.89 | 71.35 |
| | QuaRot | dynamic | 6.80 | 71.68 |
| | PrefixQuant w/o FT | static | 6.90 | 70.29 |
| | PrefixQuant | static | **6.62** | **72.46** |
| Llama-3-70B | QoQ | dynamic | 4.36 | 78.12 |
| | QuaRot | dynamic | 3.73 | **78.92** |
| | PrefixQuant w/o FT | static | 3.55 | 77.82 |
| | PrefixQuant | static | **3.43** | 78.70 |

for per-tensor quantization. In the performance comparison tables, "PrefixQunt w/o FT" indicates finishing the quantization only with grid search and without fine-tuning.

**Fine-Tuning Setting.** During fine-tuning, we optimize block output mean square error following existing works (Shao et al., 2023; Chen et al., 2024a). The dataset for fine-tuning consists of 512 samples from Pile with a 1024 context length. The learning rates for quantization parameters (step sizes) and full-precision weights are set to 5e-5 and 5e-6, respectively, and to 2e-5 and 2e-6 for Llama-3-70B(-Instruct) models. The fine-tuning batch size is set to 4, and the number of epochs is set to 10 for W4A8KV4 and 20 for W4A4KV4.

## 6.2 COMPARISON RESULTS

**Results on W4A4KV4.** Table 3 shows the comparison results for W4A4KV4. PrefixQuant with static quantization significantly outperforms the previous state-of-the-art QuaRot, which uses dynamic quantization. For instance, in Llama-3-8B, PrefixQuant without fine-tuning surpasses QuaRot by 0.48 in WikiText2 perplexity and +3.22 points in average accuracy. Fine-tuning further improves these results to 0.98 in WikiText2 perplexity and +5.98 points in average accuracy.

**Results on W4A8KV8.** Table 4 presents the comparison results for W4A8KV8. Without fine-tuning, PrefixQuant performs comparably to QoQ (Lin et al., 2024b). After fine-tuning, PrefixQuant outperforms both QoQ and QuaRot in most models. For instance, PrefixQuant surpasses QoQ (Lin et al., 2024b) by 0.27 perplexity and +1.11 accuracy points in Llama-3-8B.

**Results on W8A8KV8.** Table 18 includes the comparison with various methods in W8A8KV8 quantization. We can find that SmoothQuant, QuaRot, and PrefixQuant all attain near lossless performance without fine-tuning. Notably, our PrefixQuant is unique in employing static quantization, which enhances inference efficiency. Additionally, earlier methods like CushionCache (Son et al., 2024) and QFeP (Yang et al., 2024), despite also using prefixed tokens in the KV cache to support coarser quantization, exhibit marked performance decline even under W8A8 as illustrated in Table 17.

**Results on more models.** The results in Table 19 demonstrate that PrefixQuant consistently achieves excellent performance on other models such as Mistral-7b-v0.3 and Qwen-2-7B, as well as instruction-tuned models like Llama-3-{7B,70B}-Instruct.

**Results on weight-only quantization.** PrefixQuant can also improve existing weight-only quantization methods by reducing outlier noise in MSE loss calculations. As shown in Table 16, PrefixQuant

enhances the average accuracy by $+5.05$ and $+4.73$ points on W2A16g128 Llama-3-8B and Llama-3-70B, respectively, based on the state-of-the-art uniform quantization method EfficientQAT (Chen et al., 2024a). See Sec. G for more details.

## 6.3 INFERENCE SPEED

In this section, we evaluate the end-to-end inference speed of PrefixQuant in the W4A4 quantization scenario. We do not consider KV quantization here because it saves memory footprint through more computation overhead and only achieves speedup with large batch sizes (Liu et al., 2024b). Table 5 shows the speedup of W4A4 quantization in the prefilling stage. Our PrefixQuant improves the QuaRot speedup from $2.30\times$ to $2.81\times$ on the A100-80GB GPU, and from $1.31\times \sim 1.39\times$ to $1.60\times \sim 1.82\times$ on the RTX 3090 GPU. In Sec. D, we also provide comprehensive apple-to-apple comparisons of submodules, such as quantization kernels and quantized linears, demonstrating the significant superiority of PrefixQuant over the existing dynamic quantization approach QuaRot (Ashkboos et al., 2024b).

Table 5: Time-to-first-token (prefilling) speedup of W4A4 Llama-3-8B model over the FP16 model. We use 2048 sequence length with different batch size.

| Batchsize | 1 | 4 |
|---|---|---|
| on a RTX 3090 GPU (ms) | | |
| FP16 | 509 | OOM |
| Quarot (W4A4) | 221 (2.30x) | 851 |
| PrefixQuant (W4A4) | 181 (**2.81x**) | **725** |
| on an A100-80GB GPU (ms) | | |
| FP16 | 172 | 661 |
| Quarot (W4A4) | 130 (1.31x) | 477 (1.39x) |
| PrefixQuant (W4A4) | 107 (**1.60x**) | 362 (**1.82x**) |

Table 6: Ablation study on quantization techniques used in PrefixQuant. The model used here is Llama-3-8B, and WikiText2 perplexity with 2048 context length is reported.

| Method | Activation Quant. | W8A8KV8 | W4A8KV4 | W4A4KV4 |
|---|---|---|---|---|
| QuaRot | dynamic | 6.17 | 6.75 | 8.33 |
| RTN | dynamic | 6.26 | 12.66 | 1282.34 |
| + rotation | dynamic | 6.17 | 10.88 | 24.98 |
| + grid search | dynamic | 6.17 | 8.91 | 16.47 |
| + static quantization | static | 7.27 | 29.07 | 141.02 |
| **+ prefixed outliers** | **static** | **6.17** | **6.90** | **7.93** |
| **+ block-wise fine-tuning** | **static** | **6.17** | **6.63** | **7.41** |

## 6.4 ABLATION STUDIES

We examine the impact of various quantization techniques implemented in PrefixQuant. Our analysis starts with W4A4KV4 round-to-nearest (RTN) quantization on Llama-3-8B, involving per-channel weight quantization, per-token dynamic activation quantization, and per-head dynamic KV-cache quantization. We apply different techniques step-by-step and report the WikiText2 perplexity in Table 6. We find that both Hadamard rotation and grid search initialization improve performance. Then, perplexity collapses due to static quantization of activation and KV cache, but introducing prefixed outliers significantly recovers performance, even surpassing results before introducing static quantization. These benefits arise not only by reducing information loss from outlier tokens but also by helping to find accurate quantization parameters in grid searches through isolating extremely large outliers ($> 1e3$) in activation. Additionally, block-wise fine-tuning improves performance except on W8A8KV8, which is nearly lossless without fine-tuning. More ablation results related to the training dataset, training epochs, dynamic quantization, the number of prefixed tokens, and the content of prefixed tokens are in Sec. F in the Appendix.

## 7 CONCLUSION

We propose PrefixQuant, which enables static quantization to outperform dynamic quantization by effectively handling token-wise outliers through a novel prefixing approach. This technique also stabilizes model training, making it a plug-and-play module that enhances the performance of other optimization-based methods. The simplicity and broad applicability of PrefixQuant make it a promising direction for future LLM compression and optimization research.

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

OVERVIEW OF APPENDIX

We detailed the content of Appendix here:

- Sec A gives the reproducibility statement to summarize the information related to the reproduction of our method.
- Sec B details the quantization configuration and data sources of comparison methods.
- Sec. C illustrates the detailed image of hadamaed rotation within a transformer block.
- Sec. D presents the apple-to-apple sub-module comparisons of PrefixQuant and QuaRot.
- Sec. E details the quantization time of PrefixQuant.
- Sec. F gives more ablation studies of PrefixQuant, including the fine-tuning dataset, training epoch, dynamic quantiztaion and number of prefixed tokens.
- Sec. G demonstrates that proposed PrefixQuant can also play as a plug-in to enhance the performance of existing weight-only quantization methods.
- Sec. H presents the detailed accuracy number of each zero-shot task, and provide more results of PrefixQuant on Mistral-v0.3-7B, Qwen-2-7B, and Llama-3-{8B,70B}-Instruct.
- Sec. I illustrate more visualization of inputs of linear layer and **Q/K/V** on more models, including Llama-3-{8B,70B}, Mistral-7B-v0.3, Qwen-2-7B.

## A    REPRODUCIBILITY STATEMENT

In this section, we summarize the necessary information to reproduce our results. First, PrefixQuant is based on Hadamard rotation, as detailed in Sec.C. Our main contribution, setting prefixed outliers, is discussed in Sec.5.1. After configuring prefixed outliers in the KV-cache, we initialize the quantization parameters using grid search. We also offer optional block-wise fine-tuning to enhance performance. Detailed setups for grid search and fine-tuning are available in Sec.6.1. Additionally, we provide the source of detailed results for each compared method in Sec.B.

## B    CONFIGURATION AND DATA SOURCES OF COMPARISON METHODS

**Quantization Configurations.** In this study, we establish the quantization granularity for each comparison method based on the specifications provided in the original papers. Details on these settings are given in Table 7.

Table 7: Detailed quantization setting of comparison methods. All per-group quantization set group size as 128.

| Method | Weight | Activation | KV Cache |
|---|---|---|---|
| SmoothQuant | per-channel symmetric | per-token symmetric dynamic | per-tensor symmetric static |
| Atom | per-group symmetric | per-group symmetric dynamic | per-group asymmetric dynamic |
| QoQ | per-channel asymmetric | per-token symmetric dynamic | per-group asymmetric dynamic |
| QuaRot | per-channel symmetric | per-token dynamic symmetric | per-group asymmetric dynamic |
| DuQuant | per-channel asymmetric | per-token dynamic asymmetric | per-token asymmetric dynamic |
| **PrefixQuant** | per-channel symmetric | per-tensor symmetric static | per-head symmetric static |

**Data Sources.** We compare our proposed PrefixQuant with several other methods: QuaRot (Ashkboos et al., 2024b), Atom (Zhao et al., 2023), QoQ (Lin et al., 2024b), SmoothQuant (Xiao et al., 2023a), SpinQuant (Liu et al., 2024a), and EfficientQAT (Chen et al., 2024a). The data for our comparisons either come directly from the official publications of these methods, from other papers, or from our own reproduction of the methods. The source of the data for each method is outlined as follows:

- **QuaRot**: We present the performance of QuaRot using the Pile calibration dataset. The results for Llama-2 models with W4A4KV4 come from QoQ (Lin et al., 2024b), while the rest are reproduced using the official open-source code.

- **DuQuant**: We reproduce DuQuant with Pild calibration dataset through their official open-source code. Note that we change the evaluation toolbox to lm-eval v0.4.2 for more accurate evaluation.

- **Atom**: We present the performance of Atom using the Pile calibration dataset. The results are sourced from QoQ (Lin et al., 2024b).

- **QoQ**: We present the performance of QoQ using the Pile calibration dataset. The results for Llama-2 come from QoQ (Lin et al., 2024b), and the Llama-3 results are reproduced using the official open-source code.

- **SmoothQuant**: We present the performance of SmoothQuant using the Pile calibration dataset. All results are reproduced using the open-source code from QoQ (Lin et al., 2024b).

- **SpinQuant**: All results are reproduced using the official open-source code and the pre-trained rotation matrix. Note that SpinQuant directly trains on the WikiText2 dataset.

- **EfficientQAT**: All results are reproduced using the official open-source code and the pre-quantized models.

## C    DETAILS OF ROTATION

Hadamard rotation (Ashkboos et al., 2024b; Liu et al., 2024a) redistributes outlier channels across all channels, achieving uniform distribution within each token. The Hadamard matrix $\mathbf{H}$ is an orthogonal matrix with $\mathbf{HH}^T = \mathbf{I}$, and its entries are $\{+1, -1\}$ at the same scale. Hadamard rotation can be applied to all activations and use inverse rotation on corresponding weights to maintain computational invariance (Ashkboos et al., 2024a). Specifically, the rotation includes absorbable and online rotations. As shown in Figure 6, we follow SpinQuant (Liu et al., 2024a) to set $R1$, $R2$, $R3$ and $R4$ rotations, details as follows.

**Absorbable Rotation.** Hadamard rotation of activation can be absorbed into the previous linear layer if there is no intervening non-linear operation. Thus, the rotation of input activations for q/k/v/gate/up_proj ($R_1$) and head-wise rotation for o_proj input activations ($R_2$) can be fully absorbed without adding computation during inference.

**Online Rotation.** Some rotations must be executed online, including output activations of q_proj and k_proj after RoPE (Su et al., 2024) ($R_3$), and the input activation of down_proj ($R_4$). These online rotations are efficiently implemented using the Walsh-Hadamard transform without significant overhead.

If not specifically mentioned, we activate all rotation ($R_1$, $R_2$, $R_3$ and $R_4$) in weight-activation quantization scenes, and only activate absorbable rotation ($R_1$ and $R_2$) in weight-only quantization.

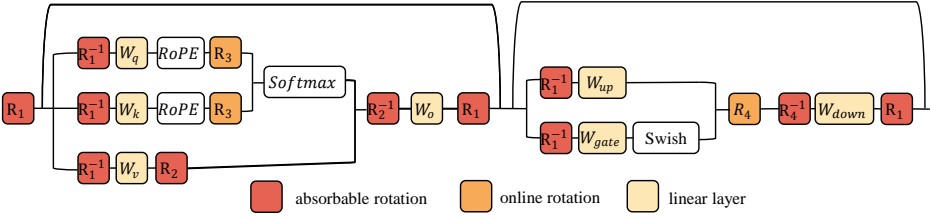

Figure 6: Illustrate of hadamard rotation within a transformer block of Llama (Touvron et al., 2023) model.

## D    INFERENCE EFFICIENCY DETAILS

Table 8: Speedup of per-tensor static quantization compared to per-token dynamic quantization in 4-bit activation quantization.

(a) **Nvidia RTX 3090 GPU**

| (Seq_len,dimension) | Quantization Time (ms) | | Speedup |
|---|---|---|---|
| | **Per-token Dynamic** | **Per-tensor static** | |
| (1,4096) | 0.0127 | 0.0038 | 3.34x |
| (1,8192) | 0.0144 | 0.004 | 3.60x |
| (2048,4096) | 0.1073 | 0.0344 | 3.12x |
| (2048,8192) | 0.2084 | 0.0652 | 3.19x |
| **Average Speedup** | | | **3.31x** |

(b) **Nvidia A100-80GB GPU**

| (Seq_len,dimension) | Quantization Time (ms) | | Speedup |
|---|---|---|---|
| | **Per-token Dynamic** | **Per-tensor static** | |
| (1,4096) | 0.020 | 0.0072 | 2.81x |
| (1,8192) | 0.022 | 0.0075 | 2.88x |
| (2048,4096) | 0.095 | 0.033 | 2.88x |
| (2048,8192) | 0.157 | 0.058 | 2.71x |
| **Average Speedup** | | | **2.82x** |

Table 9: Speedup of W4A4 quantized linear layers compared to FP16 linear layer. Numbers in brackets indicate the speedup compared to FP16.

(a) **Nvidia RTX 3090 GPU**

| (Seq_len,input_c, output_c) | FP16 (ms) | W4A4 (ms) | | |
|---|---|---|---|---|
| | | Quarot | + static quant | + improved GEMV |
| (1,4096,4096) | 0.0512 | 0.0578 (0.89x) | 0.0472 (1.08x) | 0.0223 (**2.30x**) |
| (1,4096,14336) | 0.1548 | 0.0641 (2.42x) | 0.0549 (2.83x) | 0.0475 (**3.27x**) |
| (1,8192,8192) | 0.1080 | 0.0957 (1.77x) | 0.0863 (1.97x) | 0.0561 (**3.02x**) |
| (1,8192,28672) | 0.5762 | 0.2087 (2.76x) | 0.1977 (2.91x) | 0.1503 (**3.83x**) |
| (2048,4096,4096) | 1.0666 | 0.3699 (2.88x) | 0.2965 (**3.59x**) | - |
| (2048,4096,14336) | 3.5766 | 0.9358 (3.93x) | 0.8618 (**4.27x**) | - |
| (2048,8192,8192) | 3.9986 | 1.0211 (4.03x) | 0.8718 (**4.72x**) | - |
| (2048,8192,28672) | 13.1607 | 2.8609 (4.74x) | 2.7177 (**4.99x**) | - |

(b) **Nvidia A100-80GB GPU**

| (Seq_len,input_c, output_c) | FP16 (ms) | INT4 (ms) | | |
|---|---|---|---|---|
| | | Quarot | + static quant | + improved GEMV |
| (1,4096,4096) | 0.0418 | 0.0588 (0.71x) | 0.0455 (0.92x) | 0.0235 (**1.78x**) |
| (1,4096,14336) | 0.1026 | 0.0679 (1.51x) | 0.0556 (1.85x) | 0.0441 (**2.33x**) |
| (1,8192,8192) | 0.1080 | 0.0888 (1.22x) | 0.0735 (1.47x) | 0.0508 (**2.13x**) |
| (1,8192,28672) | 0.3036 | 0.1668 (1.82x) | 0.1534 (1.97x) | 0.1114 (**2.72x**) |
| (2048,4096,4096) | 0.2762 | 0.2408 (1.15x) | 0.1799 (**1.54x**) | - |
| (2048,4096,14336) | 1.0092 | 0.5461 (1.85x) | 0.4850 (**2.08x**) | - |
| (2048,8192,8192) | 1.0583 | 0.5298 (2.00x) | 0.4349 (**2.43x**) | - |
| (2048,8192,28672) | 3.6897 | 1.4686 (2.51x) | 1.3857 (**2.66x**) | - |

In this section, we examine the inference efficiency of PrefixQuant. We conduct tests on Nvidia RTX 3090 and A100-80GB GPUs, considering sequence lengths of 1 and 2048, with a batch size of

1. We detail the speedup ratios for quantization overhead, quantized linear layers, and end-to-end inference below.

**Reduced Quantization Overhead.** Activations are quantized and packed into low-bit formats for matrix manipulations. We define the time for this process during inference as quantization overhead. The per-token dynamic quantization kernel is sourced from QuaRot (Ashkboos et al., 2024b). Table 8 shows the speedup of per-tensor static quantization over per-token dynamic quantization. We can find that the per-tensor static quantization kernel achieves speedups of $3.31\times$ on the RTX 3090 and $2.82\times$ on the A100-80GB.

**Accelerated Quantized Linear Layer.** The quantized linear layer consists of quantization, low-bit matrix multiplication, and de-quantization. The speedup for the quantization process is in Table 8. For low-bit matrix multiplication, we use the 4-bit GEMM kernel from CUTLASS and design a custom kernel for W4A4 GEMV. We also integrate the de-quantization process into the GEMM and GEMV kernels. Table 9 presents the speedup ratios of the QuaRot kernel and our kernel compared to FP16. With a sequence length of 1, our quantized linear layer improves the QuaRot speedup from $0.89\times \sim 2.76\times$ to $2.30\times \sim 3.83\times$ on the RTX 3090, and from $0.71\times \sim 1.82\times$ to $1.78\times \sim 2.72\times$ on the A100-80GB. With a sequence length of 2048, our layer enhances the QuaRot speedup from $2.88\times \sim 4.74\times$ to $3.59\times \sim 4.99\times$ on the RTX 3090, and from $1.15\times \sim 2.51\times$ to $1.54\times \sim 2.66\times$ on the A100-80GB.

# E  QUANTIZATION TIME

Table 10 shows the quantization time for PrefixQuant. PrefixQuant identifies prefixed tokens quickly, taking only 0.2 minutes for Llama-3-8B and 1 minute for Llama-3-70B. In contrast, the recent CushionCache (Son et al., 2024) requires 12 hours for the same task on Llama-3-8B. Additionally, the grid-search initialization is efficient, taking 0.7 minutes for Llama-3-8B and 12 minutes for Llama-3-70B. Experiments in Tables 3 and 4 demonstrate that PrefixQuant, even without fine-tuning, outperforms previous methods (Lin et al., 2024b; Ashkboos et al., 2024b). Fine-tuning requires more time, taking 2.2 hours for Llama-3-8B and 17 hours for Llama-3-70B, but it can successfully enhances the potential of low-bit quantization.

Table 10: The quantization time of PrefixQuant on single NVIDIA-A100-80GB GPU. Fine-tuning indicates the time of 20 fine-tuning epochs of W4A4KV4.

| Model | Find Prefixed Outliers | Grid-search initialization | Fine-tuning |
|-------|------------------------|----------------------------|-------------|
| Llama-3-8B | 0.2 m | 0.7 m | 2.2 h |
| Llama-3-70B | 1 m | 12 m | 17 h |

# F  MORE ABLATION RESULTS

Table 11: Ablation studies on calibration dataset, including (a) Dataset type, (b) Training sequence length and (c) Total training tokens. "N" indicates number of training samples, and "S" is the length of each samples. The model used here is Llama-3-8B with W4A4KV4 quantization. Our default settings are marked in gray .

| (a) Dataset | | (b) Sequence length | | (c) Total token number | |
|---|---|---|---|---|---|
| **Dataset** | **Wiki PPL** | **N × S** | **Wiki PPL** | **N × S** | **Wiki PPL** |
| C4 | 7.60 | 256 ×2048 | 7.65 | 256 ×1024 | 7.46 |
| RedPajama | 7.49 | 512×1024 | **7.42** | 512 ×1024 | 7.42 |
| Pile | **7.42** | 1024×512 | 7.65 | 1024×1024 | **7.41** |

**Fine-tuning Datasets.** Table 11a shows results with different fine-tuning datasets, including C4 (Raffel et al., 2020), RedPajama (Computer, 2023), and Pile (Gao et al., 2020). We find that Pile

Table 12: Ablation study about training epochs. The model used here is Llama-3-8B, and WikiText2 perplexity with 2048 context length is reported. Our default settings are marked in  gray .

| Epochs | W4A8KV4 | W4A4KV4 |
|---|---|---|
| 0 (w/o FT) | 6.90 | 7.93 |
| 5 | 6.66 | 7.53 |
| 10 | **6.63** | 7.47 |
| 20 | 6.63 | 7.42 |
| 30 | 6.63 | **7.41** |

Table 13: Ablation study about quantization type of activation. The model used here is Llama-3-8B with W4A4KV4 quantization. WikiText2 perplexity with 2048 context length is reported.

| Fine-Tuning | Quant Type | W4A8KV4 | W4A4KV4 |
|---|---|---|---|
| No | token-wise dynamic | **6.84** | 8.29 |
| No | tensor-wise static | 6.90 | **7.93** |
| Yes | token-wise dynamic | **6.60** | 7.88 |
| Yes | tensor-wise static | 6.63 | **7.41** |

Table 14: Ablation study about the number of prefixed tokens. WikiText2 perplexity with 2048 context length and W4A4KV4 quantization is reported. Number $n$ indicates the first $n$ tokens in Table 1 are set as the prefixed tokens.

| Model | Method | 0 | 1 | 2 | 3 | 4 |
|---|---|---|---|---|---|---|
| Llama-2-7B | PrefixQuant w/o FT | 333.52 | 74.37 | 6.21 | 6.22 | - |
| Llama-2-7B | PrefixQuant | 17.63 | 10.71 | **6.01** | 6.01 | |
| Mistral-7B-v0.3 | PrefixQuant w/o FT | 90.02 | 6.12 | 5.84 | 6.43 | 5.89 |
| Mistral-7B-v0.3 | PrefixQuant | 15.97 | 7.08 | 5.83 | 5.95 | **5.79** |

achieves the best performance. Additionally, we ablate the sequence length of each training sample and the total training tokens. Table 11b shows that a sequence length of 1024 achieves the best performance. Table 11c demonstrates that fine-tuning on $512 \times 1024$ tokens achieves satisfactory performance, with further increases in training samples only marginally improving performance. Note that the optimal token number for fine-tuning datasets may change with quantization precision. Generally, lower precision requires more training data. For example, EfficientQAT shows that $4096 \times 2048$ tokens are needed for W2A16 quantization, while our paper shows that only $512 \times 1024$ tokens are needed for W4A4 quantization.

**Training Epochs.** Table 12 demonstrates that 10 and 20 epochs are sufficient for the convergence of fine-tuning on W4A8KV4 and W4A4KV4.

**Dynamic Quantization.** Tables 3 and 4 show that PrefixQuant with static quantization can surpass previous state-of-the-art methods (Xiao et al., 2023a; Ashkboos et al., 2024b; Lin et al., 2024b) with dynamic quantization. Note that without prefixed outliers, per-token dynamic quantization consistently outperforms per-tensor static quantization across different precisions, as shown in Table 6. Therefore, a question arises: can dynamic quantization further improve the performance of PrefixQuant? We replace per-tensor static activation quantization in PrefixQuant with per-token dynamic quantization and report the results in Table 13. We find that the winner varies with different precision. Specifically, per-token dynamic quantization marginally surpasses per-tensor static quantization in W4A8KV4 quantization, while per-tensor static quantization significantly outperforms per-token dynamic quantization in W4A4KV4 quantization. This is because, in high-precision quantization such as 8-bit, clipping is not necessary (Gong et al., 2024), and the MAX-MIN initialization of dynamic quantization adapts to a more diverse range flexibly. However, in low-precision quantization such as 4-bit, clipping is crucial to balance clipping error and rounding error (Lin et al., 2023), resulting in per-tensor static quantization outperforming per-token dynamic quantization.

Table 15: Ablation study about the content of prefixed tokens. WikiText2 perplexity with 2048 context length and W4A4KV4 quantization is reported. "default" refers to the prefixed tokens obtained through the proposed method. "random" represents the average performance of 10 times with randomly selected prefixed tokens.

| Model | Type | Prefixed | Wiki PPL (PrefixQuant w/o FT) |
|---|---|---|---|
| Llama-2-7B | default | .\n[BOS] | 6.22 |
| Llama-2-7B | only highest frequency | ... | 12.07 |
| Llama-2-7B | random | - | 66.51 |
| Mistral-7B-v0.3 | default | \n.to[BOS] | 5.89 |
| Mistral-7B-v0.3 | only highest frequency | \n\n\n\n | 6.23 |
| Mistral-7B-v0.3 | random | - | 80.05 |

**Number of Prefixed Tokens.** In Sec. 5.1, we determine the number of prefixed tokens by calculating the average number of outlier tokens and adding an additional [BOS] token. Table 1 illustrates the specific number and content of these tokens. We use Llama-2-7B (3 outlier tokens) and Mistral-7B-v0.3 (4 outlier tokens) to study the impact of the number of prefixed tokens. Table 14 shows that the adaptively calculated number of prefixed tokens achieves the best performance. Notably, for models like Llama-2-7B, using 2 prefixed tokens without the additional [BOS] token also yields excellent performance. For consistency and simplicity, we include the [BOS] token in the prefixed tokens in our experiments.

**Content of Prefixed Tokens.** PrefixQuant determines the number of outlier tokens $N$ and designates the top-$N$ high-frequency outlier tokens as prefixes in the KV cache. Table 15 examines various prefixed tokens with the same token count. The results show that using the top-$N$ high-frequency tokens as prefixed tokens significantly outperforms using only the highest-frequency or randomly selected tokens.

Table 16: Weight-only quantization results. "g" indicates group size for weight quantization. EfficientQAT only execute Block-AP and without E2E-QP for the fair comparisons in block-wise reconstruction scenario. We providing WikiText2 perplexity with 2048 context length and detailed zero-shot accuracy of weight-only quantization by `lm_eval v0.4.2`. We report `acc` for WinoGrande and `acc_norm` for HellaSwag, ArcC, ArcE, and PIQA.

| Model | Method | Precision | Wiki PPL | WinoGrande | HellaSwag | ArcC | ArcE | PiQA | Avg. Acc. |
|---|---|---|---|---|---|---|---|---|---|
| | Baseline | FP16 | 6.14 | 72.61 | 79.17 | 53.41 | 77.69 | 80.69 | 72.71 |
| | EfficientQAT | W3A16g128 | 7.34 | 70.48 | 75.09 | 51.37 | 77.9 | 79.16 | 70.80 |
| 3-8B | PrefixQuant | W3A16g128 | 7.17 | 72.38 | 76.54 | 52.65 | 78.37 | 80.58 | 72.10 |
| | EfficientQAT | W2A16g128 | 13.55 | 62.04 | 62.49 | 36.6 | 60.44 | 73.18 | 58.95 |
| | PrefixQuant | W2A16g128 | 11.97 | 66.22 | 66.54 | 41.81 | 69.61 | 75.84 | 64.00 |
| | Baseline | FP16 | 2.85 | 80.51 | 84.9 | 64.33 | 85.9 | 84.49 | 80.03 |
| | EfficientQAT | W3A16g128 | 4.89 | 78.77 | 83.74 | 55.03 | 78.66 | 82.05 | 75.65 |
| 3-70B | PrefixQuant | W3A16g128 | 4.79 | 78.22 | 84.03 | 60.15 | 83.00 | 83.35 | 77.75 |
| | EfficientQAT | W2A16g128 | 16.79 | 66.14 | 73.01 | 48.21 | 73.57 | 78.45 | 67.88 |
| | PrefixQuant | W2A16g128 | 11.01 | 72.3 | 78.55 | 53.67 | 77.9 | 80.63 | 72.61 |

# G  EXTEND TO WEIGHT-ONLY QUANTIZATION

In addition to static activation quantization, setting prefixed outliers in the KV-cache improves training stability (Chen et al., 2024b) and reduces information loss from outlier tokens, can also enhancing weight-only quantization performance. To verify this, we compare PrefixQuant with the recent state-of-the-art weight-only quantization method, EfficientQAT (Chen et al., 2024a), in a block-wise fine-tuning scenario. Following EfficientQAT, we use 4096 RedPajama (Computer, 2023) with a 2048 context length to train for 2 epochs. The learning rates for quantization parameters and full-precision weights are set to 5e-5 and 5e-6, except for W2A16g128 Llama-3-8B, where they are 1e-4 and 2e-5, respectively. As shown in Table 16, PrefixQuant significantly surpasses EfficientQAT

with $+5.05$ and $+4.73$ points in average accuracy on W2A16g128 Llama-3-8B and Llama-3-70B, respectively.

## H    FULL RESULTS OF WEIGHT-ACTIVATION QUANTIZATION

Table 17: W8A8 performance comparisons with other methods that also set prefixed tokens in KV cache.

| Model | Method | Activation Quant | Wiki PPL |
|---|---|---|---|
| 2-7B | QFeP | per-tensor dynamic | 5.75 |
| | CushionCache | per-tensor static | 5.87 |
| | PrefixQuant | per-tensor static | **5.48** |
| 2-13B | QFeP | per-tensor dynamic | 6.00 |
| | PrefixQuant | per-tensor static | **4.89** |
| 2-70B | QFeP | per-tensor dynamic | 6.01 |
| | PrefixQuant | per-tensor static | **3.39** |
| 3-8B | CushionCache | per-tensor static | 7.37 |
| | PrefixQuant | per-tensor static | **6.17** |

### H.1    COMPARISONS WITH RELATED WORKS

CushionCache (Son et al., 2024) and QFeP (Yang et al., 2024) also set prefixed tokens in the KV cache to reduce outliers. However, they experience significant performance degradation even with W8A8 quantization. Table 17 shows that PrefixQuant outperforms QFeP by 2.62 perplexity on Llama-2-70B and surpasses CushionCache by 1.20 perplexity on Llama-3-8B.

### H.2    DETAILED ACCURACY RESULTS

In the main paper, we present the average accuracy of five common reasoning tasks for brevity. Here, we provide detailed results for each task in Table 18.

### H.3    RESULTS ON MORE MODELS

Table 19 shows the effectiveness of the proposed PrefixQuant in other models, including Mistral-v0.3-7B and Qwen-2-7B. It also includes instruction-tuned models such as Llama-3-{8B,70B}-Instruct.

## I    MORE VISUALIZATIONS

### I.1    OUTLIER TOKEN

In Figure 7, we showcase the four most frequently occurring outlier tokens in Llama-2-{13B,70B}, Llama-3-70B, and Mistral-7B-v0.3. Specifically, Table 1 selects the top-$o$ high-frequent outlier tokens as the prefixed tokens. It is important to note that we do not visualize the outlier tokens in Llama-3-8B and Qwen-2-7B because all the outlier tokens in these two models appear in the initial tokens.

### I.2    MAGNITUDE DISTRIBUTION

We illustrate more token-wise maximum values distribution of other models. Details are as follows:

- **Llama-2-13B**: Figure 8 and Figure 9 illustrate the distribution of input activation and **Q/K/V**, respectively.

Table 18: Continuation of Table 3 and Table 4, providing detailed zero-shot accuracy of weight-activation quantization of Llama models by `lm_eval v0.4.2`. We report `acc` for WinoGrande and `acc_norm` for HellaSwag, ArcC, ArcE, and PIQA.).

| Model | Method | Precision | WinoGrande | HellaSwag | ArcC | ArcE | PiQA | Avg. Acc. |
|-------|--------|-----------|-----------|-----------|------|------|------|-----------|
| 2-7B | Baseline | FP16 | 69.22 | 76.00 | 46.25 | 74.62 | 79.11 | 69.04 |
| | Atom | W4A4KV4 | 62.75 | 69.37 | 38.40 | 52.99 | 75.14 | 59.73 |
| | QuaRot | W4A4KV4 | 64.40 | 72.3 | 41.47 | 68.06 | 76.17 | 64.48 |
| | DuQuant | W4A4KV4 | 67.09 | 72.53 | 43.26 | 71.38 | 76.99 | 66.25 |
| | SpinQuant | W4A4KV4 | 66.54 | 73.15 | 41.64 | 69.32 | 76.12 | 65.35 |
| | PrefixQuant w/o FT | W4A4KV4 | 67.80 | 73.75 | 43.94 | 71.51 | 77.2 | 66.84 |
| | PrefixQuant | W4A4KV4 | 66.54 | 73.42 | 43.09 | 71.17 | 77.64 | 66.37 |
| | QoQ | W4A8KV8 | 68.03 | 74.00 | 43.60 | 72.81 | 77.64 | 67.22 |
| | QuaRot | W4A8KV8 | 66.77 | 74.56 | 43.86 | 72.39 | 77.97 | 67.11 |
| | PrefixQuant w/o FT | W4A8KV8 | 69.14 | 75.12 | 44.45 | 73.06 | 77.53 | 67.86 |
| | PrefixQuant | W4A8KV8 | 69.06 | 75.25 | 44.8 | 73.19 | 78.13 | 68.09 |
| | SmoothQuant | W8A8KV8 | 69.22 | 76.32 | 45.56 | 74.71 | 78.78 | 68.92 |
| | QuaRot | W8A8KV8 | 68.98 | 75.96 | 46.59 | 74.41 | 79.11 | 69.01 |
| | PrefixQuant w/o FT | W8A8KV8 | 70.48 | 76.62 | 45.65 | 73.91 | 78.18 | 68.97 |
| 2-13B | Baseline | FP16 | 72.22 | 79.37 | 49.06 | 77.48 | 80.52 | 71.73 |
| | Atom | W4A4KV4 | 67.40 | 73.84 | 42.32 | 57.49 | 76.50 | 63.51 |
| | QuaRot | W4A4KV4 | 67.88 | 75.28 | 45.65 | 72.35 | 77.48 | 67.73 |
| | DuQuant | W4A4KV4 | 68.9 | 76.65 | 47.7 | 74.24 | 78.18 | 69.13 |
| | SpinQuant | W4A4KV4 | 67.88 | 77.01 | 46.76 | 75.97 | 78.56 | 69.24 |
| | PrefixQuant w/o FT | W4A4KV4 | 72.06 | 76.54 | 46.67 | 75.8 | 78.51 | 69.92 |
| | PrefixQuant | W4A4KV4 | 72.53 | 76.12 | 47.70 | 76.09 | 79.38 | 70.36 |
| | QoQ | W4A8KV8 | 70.96 | 77.80 | 48.38 | 75.97 | 79.71 | 70.56 |
| | QuaRot | W4A8KV8 | 70.24 | 78.21 | 47.01 | 74.49 | 79.87 | 69.96 |
| | PrefixQuant w/o FT | W4A8KV8 | 72.77 | 77.49 | 48.12 | 77.06 | 79.92 | 71.07 |
| | PrefixQuant | W4A8KV8 | 72.77 | 77.54 | 48.72 | 76.81 | 80.41 | 71.25 |
| | SmoothQuant | W8A8KV8 | 72.14 | 79.34 | 48.89 | 77.31 | 80.2 | 71.58 |
| | QuaRot | W8A8KV8 | 71.98 | 79.35 | 49.23 | 77.4 | 80.47 | 71.69 |
| | PrefixQuant w/o FT | W8A8KV8 | 72.53 | 78.38 | 48.98 | 76.81 | 80.9 | 71.52 |
| 2-70B | Baseline | FP16 | 79.48 | 84.31 | 56.91 | 80.30 | 82.54 | 76.71 |
| | Atom | W4A4KV4 | 74.27 | 79.06 | 46.08 | 58.25 | 79.92 | 67.52 |
| | QuaRot | W4A4KV4 | 76.24 | 81.82 | 56.23 | 80.43 | 82.43 | 75.43 |
| | DuQuant | W4A4KV4 | 75.45 | 81.95 | 55.03 | 79 | 82.32 | 74.75 |
| | SpinQuant | W4A4KV4 | 75.85 | 82.36 | 56.31 | 79.17 | 81.61 | 75.19 |
| | PrefixQuant w/o FT | W4A4KV4 | 75.45 | 80.51 | 52.3 | 77.06 | 81.12 | 73.29 |
| | PrefixQuant | W4A4KV4 | 77.35 | 82.3 | 56.4 | 79.29 | 82.05 | 75.48 |
| | QoQ | W4A8KV8 | 77.51 | 82.78 | 56.83 | 79.80 | 82.64 | 75.91 |
| | QuaRot | W4A8KV8 | 77.03 | 83.30 | 57.08 | 81.27 | 82.86 | 76.31 |
| | PrefixQuant w/o FT | W4A8KV8 | 77.35 | 82.79 | 54.35 | 78.28 | 82.21 | 75.00 |
| | PrefixQuant | W4A8KV8 | 79.08 | 83.56 | 57.42 | 80.39 | 82.05 | 76.50 |
| | SmoothQuant | W8A8KV8 | 77.03 | 83.38 | 56.91 | 80.72 | 82.92 | 76.19 |
| | QuaRot | W8A8KV8 | 77.82 | 83.8 | 57.34 | 80.93 | 82.75 | 76.53 |
| | PrefixQuant w/o FT | W8A8KV8 | 79.16 | 84.14 | 55.8 | 78.87 | 82.59 | 76.11 |
| 3-8B | Baseline | FP16 | 72.61 | 79.17 | 53.41 | 77.69 | 80.69 | 72.71 |
| | QuaRot | W4A4KV4 | 65.98 | 72.38 | 44.45 | 67.3 | 75.63 | 65.15 |
| | DuQuant | W4A4KV4 | 68.59 | 74.27 | 46.5 | 70.41 | 75.9 | 67.13 |
| | SpinQuant | W4A4KV4 | 69.22 | 74.83 | 45.99 | 74.07 | 77.04 | 68.23 |
| | PrefixQuant w/o FT | W4A4KV4 | 69.14 | 75.46 | 47.1 | 72.94 | 77.2 | 68.37 |
| | PrefixQuant | W4A4KV4 | 71.9 | 75.44 | 50.68 | 78.32 | 79.05 | 71.08 |
| | QoQ | W4A8KV8 | 73.4 | 77.23 | 50.87 | 75.59 | 79.65 | 71.35 |
| | QuaRot | W4A8KV8 | 72.74 | 77.35 | 51.62 | 77.48 | 79.22 | 71.68 |
| | PrefixQuant w/o FT | W4A8KV8 | 71.19 | 77.65 | 48.98 | 73.99 | 79.65 | 70.29 |
| | PrefixQuant | W4A8KV8 | 72.53 | 77.97 | 52.65 | 79.25 | 79.92 | 72.46 |
| | SmoothQuant | W8A8KV8 | 73.01 | 78.99 | 53.07 | 77.82 | 80.74 | 72.73 |
| | QuaRot | W8A8KV8 | 72.53 | 78.99 | 53.67 | 78.03 | 80.63 | 72.77 |
| | PrefixQuant w/o FT | W8A8KV8 | 74.11 | 79.25 | 53.75 | 78.03 | 80.36 | 73.10 |
| 3-70B | Baseline | FP16 | 80.51 | 84.9 | 64.33 | 85.9 | 84.49 | 80.03 |
| | QuaRot | W4A4KV4 | 68.51 | 76.75 | 47.01 | 72.31 | 77.37 | 68.39 |
| | DuQuant | W4A4KV4 | 70.8 | 79.89 | 59.04 | 82.91 | 81.83 | 74.89 |
| | SpinQuant | W4A4KV4 | 76.4 | 80.9 | 56 | 77.3 | 80.8 | 74.28 |
| | PrefixQuant w/o FT | W4A4KV4 | 77.43 | 83.48 | 58.87 | 79.88 | 82.32 | 76.40 |
| | PrefixQuant | W4A4KV4 | 77.35 | 83.79 | 60.15 | 81.31 | 83.3 | 77.18 |
| | QoQ | W4A8KV8 | 80.11 | 83.7 | 61.01 | 82.79 | 83 | 78.12 |
| | QuaRot | W4A8KV8 | 80.35 | 84.03 | 62.12 | 84.64 | 83.46 | 78.92 |
| | PrefixQuant w/o FT | W4A8KV8 | 79.23 | 84.71 | 59.39 | 81.57 | 84.22 | 77.82 |
| | PrefixQuant | W4A8KV8 | 79.48 | 84.86 | 62.29 | 82.53 | 84.33 | 78.70 |
| | SmoothQuant | W8A8KV8 | 79.40 | 84.64 | 63.14 | 85.35 | 83.9 | 79.29 |
| | QuaRot | W8A8KV8 | 80.66 | 84.84 | 63.65 | 85.56 | 84.44 | 79.83 |
| | PrefixQuant w/o FT | W8A8KV8 | 79.40 | 85.5 | 61.43 | 82.49 | 84.22 | 78.61 |

- **Llama-3-8B**: Figure 10 and Figure 11 illustrate the distribution of input activation and **Q/K/V**, respectively.

Table 19: Results of proposed PrefixQuant on other models.

| Model | Precision | Wiki PPL | WinoGrande | HellaSwag | ArcC | ArcE | PiQA | Avg. Acc. |
|---|---|---|---|---|---|---|---|---|
| Mistral-v0.3-7B | FP16 | 5.32 | 73.88 | 80.43 | 52.3 | 78.28 | 82.26 | 73.43 |
| | W8A8KV8 | 5.34 | 74.03 | 80.8 | 53.5 | 79.76 | 81.72 | 73.96 |
| | W4A8KV4 | 5.51 | 73.88 | 79.8 | 52.05 | 79.42 | 80.79 | 73.19 |
| | W4A4KV4 | 5.79 | 71.51 | 78.12 | 49.66 | 78.03 | 79.92 | 71.45 |
| Qwen-2-7B | FP16 | 7.14 | 72.3 | 78.96 | 52.65 | 78.75 | 80.96 | 72.72 |
| | W8A8KV8 | 7.15 | 72.22 | 78.88 | 52.9 | 78.49 | 80.85 | 72.67 |
| | W4A8KV4 | 8.04 | 71.43 | 76.77 | 53.67 | 77.95 | 78.45 | 71.65 |
| | W4A4KV4 | 8.37 | 68.75 | 74.92 | 48.21 | 74.75 | 79.49 | 69.22 |
| Llama-3-8B-Instruct | FP16 | 8.29 | 71.82 | 75.81 | 56.83 | 79.76 | 78.51 | 72.55 |
| | W8A8KV8 | 8.21 | 71.35 | 75.54 | 56.31 | 78.75 | 79.16 | 72.22 |
| | W4A8KV4 | 8.74 | 70.17 | 74.6 | 54.44 | 77.65 | 77.97 | 70.97 |
| | W4A4KV4 | 8.96 | 69.53 | 74.66 | 52.65 | 76.35 | 76.66 | 69.97 |
| Llama-3-70B-Instruct | FP16 | 5.33 | 75.69 | 82.58 | 64.42 | 84.97 | 82.15 | 77.96 |
| | W8A8KV8 | 5.40 | 78.06 | 82.67 | 66.72 | 84.89 | 82.21 | 78.91 |
| | W4A8KV4 | 5.96 | 77.74 | 81.97 | 65.87 | 84.93 | 81.56 | 78.41 |
| | W4A4KV4 | 6.80 | 75.93 | 80.64 | 64.76 | 83.88 | 81.23 | 77.29 |

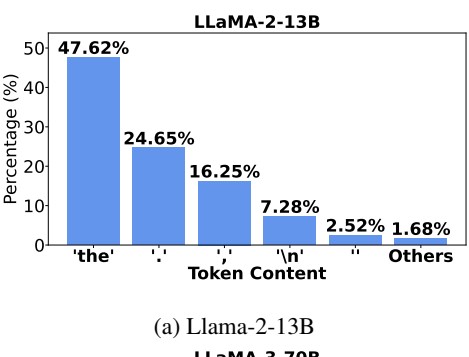

(a) Llama-2-13B

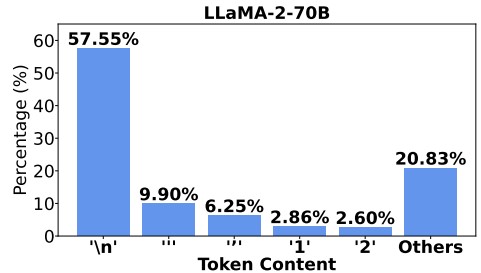

(b) Llama-2-70B

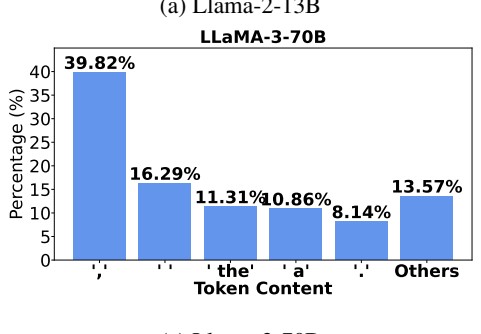

(c) Llama-3-70B

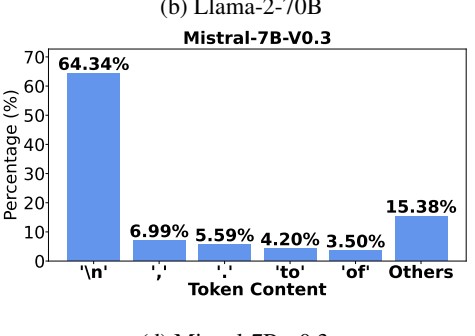

(d) Mistral-7B-v0.3

Figure 7: **Content of outlier tokens in different models.** Note that we do not count the outlier tokens situated at the initial token.

- **Llama-3-70B**: Figure 12 and Figure 13 illustrate the distribution of input activation and **Q/K/V**, respectively.

- **Qwen-2-7B**: Figure 14 and Figure 15 illustrate the distribution of input activation and **Q/K/V**, respectively.

- **Mistral-7B-v0.3**: Figure 16 and Figure 17 illustrate the distribution of input activation and **Q/K/V**, respectively.

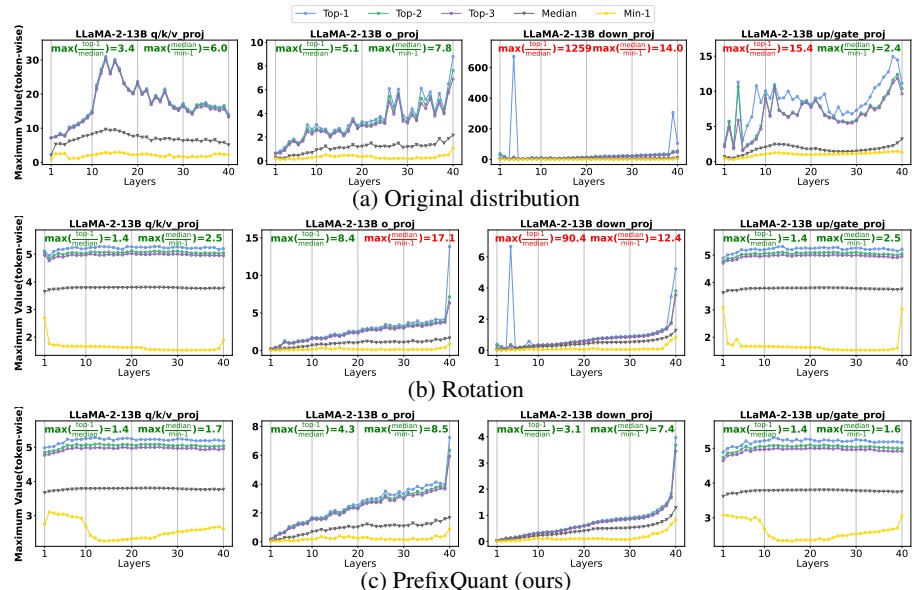

Figure 8: **Distribution of token-wise maximum values for linear layers inputs in Llama-2-13b.**

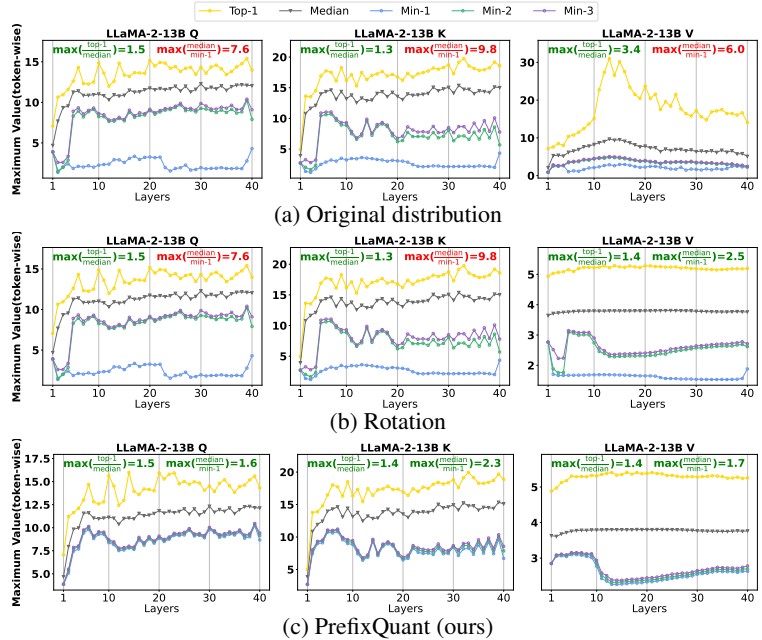

Figure 9: **Distribution of token-wise maximum values for Q/K/V in Llama-2-13b.** Same present rules as Figure 8a except that ratios greater than 5 are marked with red.

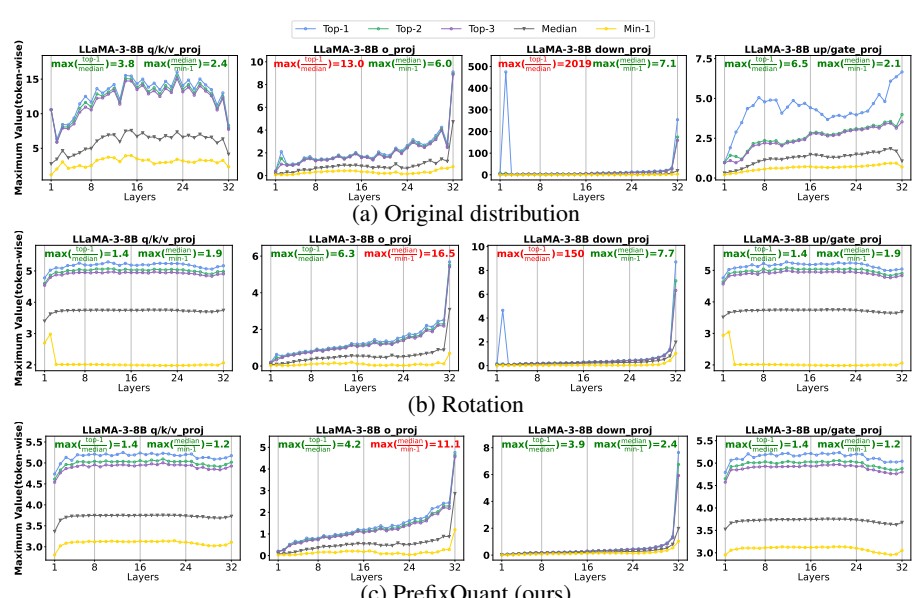

Figure 10: **Distribution of token-wise maximum values for linear layers inputs in Llama-3-8b.**

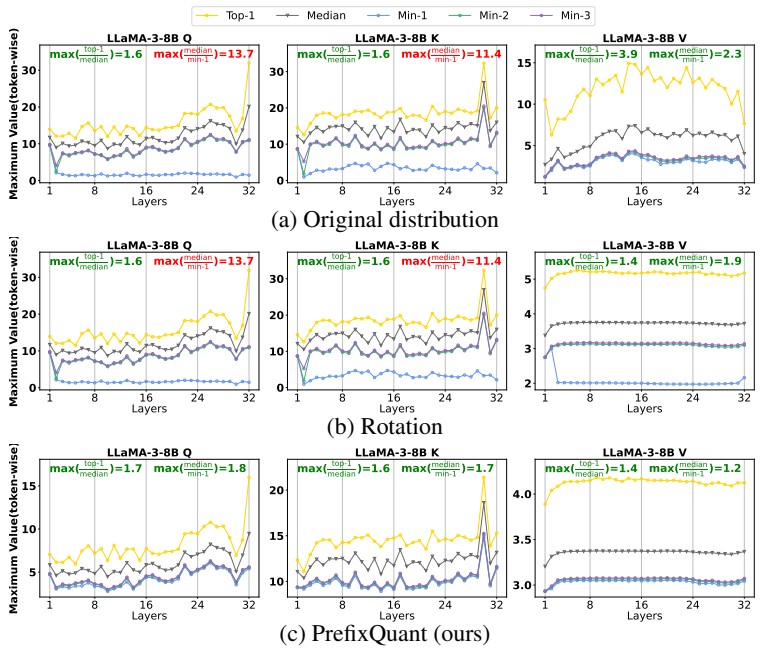

Figure 11: **Distribution of token-wise maximum values for Q/K/V in Llama-3-8B.**

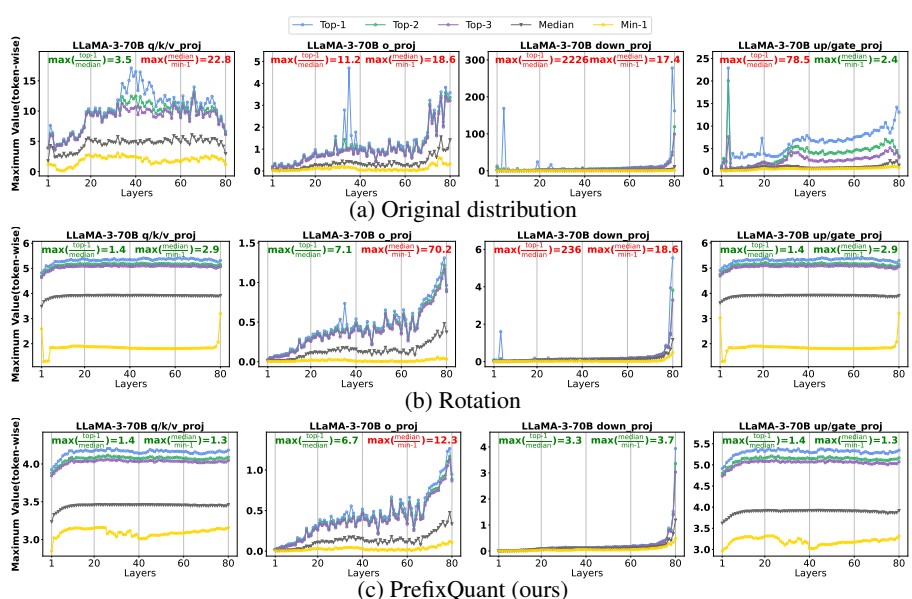

Figure 12: **Distribution of token-wise maximum values for linear layers inputs in Llama-3-70B.**

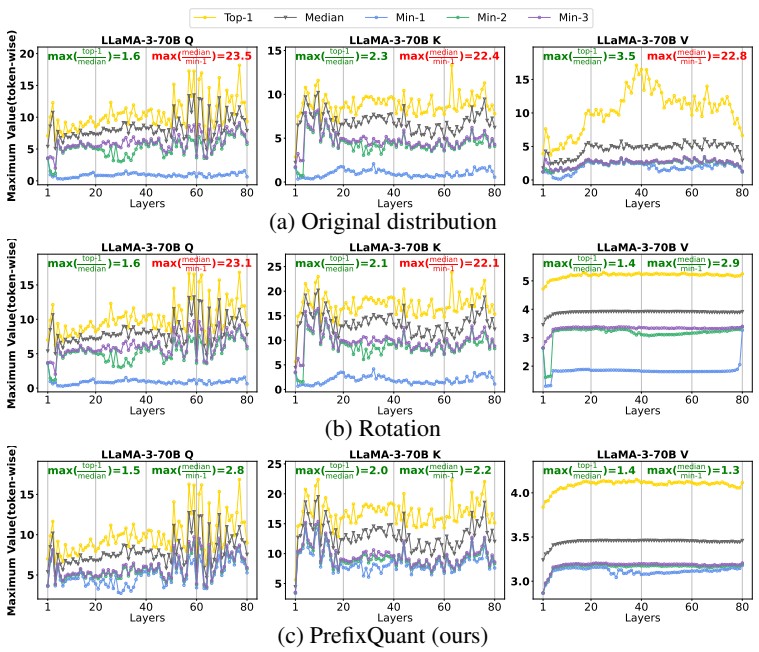

Figure 13: **Distribution of token-wise maximum values for Q/K/V in Llama-3-70B.**

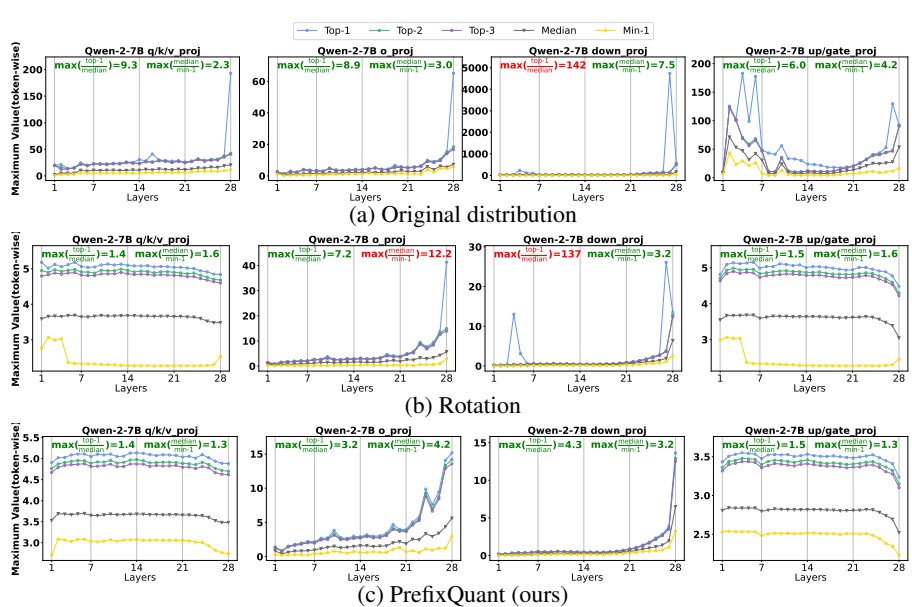

Figure 14: **Distribution of token-wise maximum values for linear layers inputs in Qwen-2-7B.**

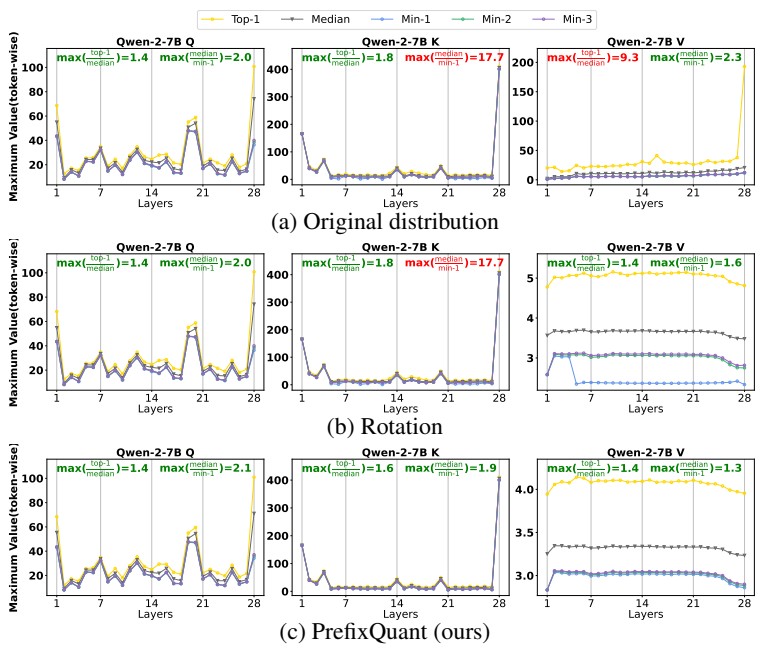

Figure 15: **Distribution of token-wise maximum values for Q/K/V in Qwen-2-7B.**

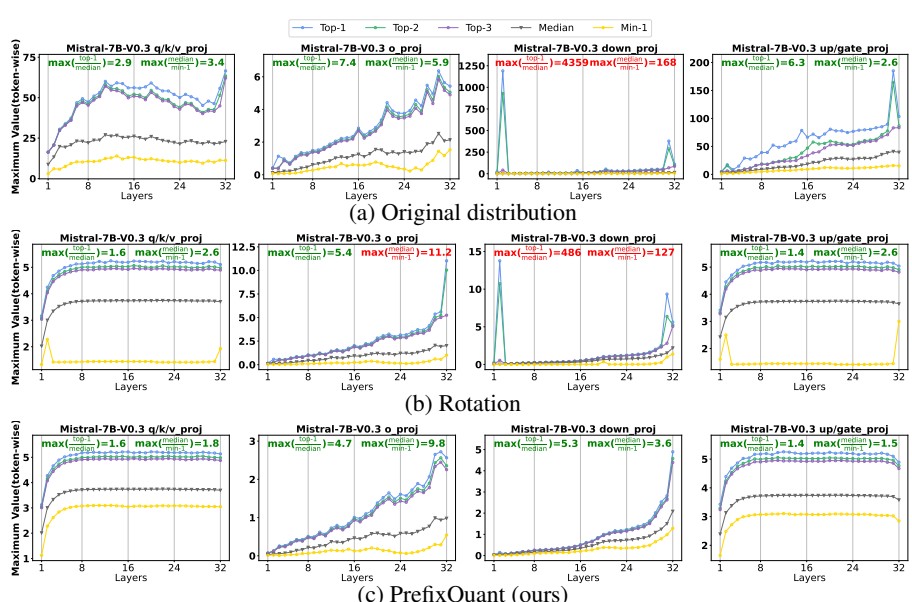

Figure 16: **Distribution of token-wise maximum values for linear layers inputs in Mistral-7B-v0.3.**

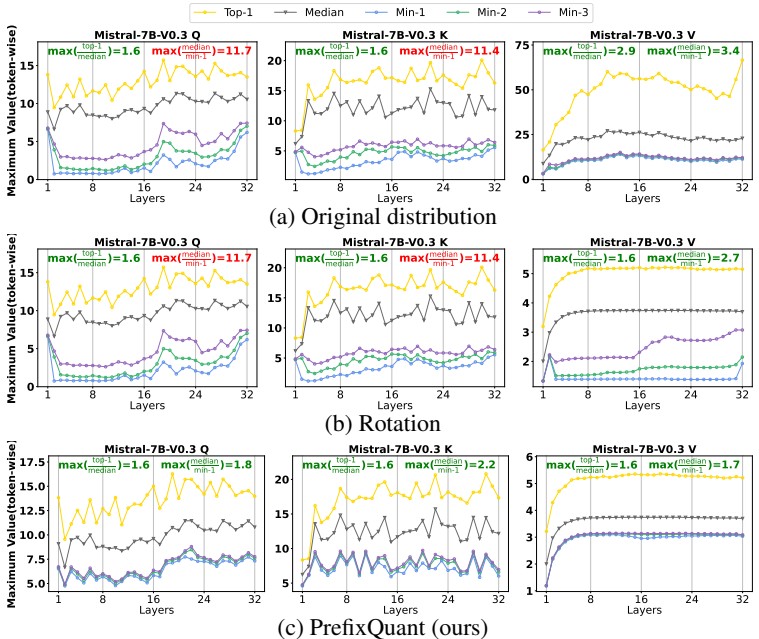

Figure 17: **Distribution of token-wise maximum values for Q/K/V in Mistral-7b-v0.3.**

