# OpenReview forum: "PrefixQuant: Static Quantization Beats Dynamic through Prefixed Outliers in LLMs"
_ICLR.cc/2025/Conference — Submitted to ICLR 2025_

### Official Review · Reviewer_wZKZ · 2024-11-02

**Soundness:** 2
**Presentation:** 1
**Contribution:** 3
**Rating:** 3
**Confidence:** 3

**Summary:**

From a high-level, I understand the idea of the paper is to identify several tokens that have high outlier values from a calibration set, and then prefix these values ahead of the time in the KV cache. The author then used some empirical measure (max over median) to obtain these outlier tokens. At inference time, the outlier tokens are somehow skipped so that there is a less profound outlier effect in the activations, and thus make the whole flow more quantization friendly so that the authors can apply a static quantization in which we do not have to quantize and dequantize at run-time. However, I think some technical detail is either missing or not carefully explained, making it hard to understand how the proposed benefits on quantization is materialized.

**Strengths:**

The paper describes a novel method that deals with the known outlier problem for quantization on the token dimension. The proposed method carries simplicity and novelty in its current description, and also has a low-cost when executing it in practise.

**Weaknesses:**

1. It is not super clear how the proposed method work, specially, I do not really understand why prefix certain outlier tokens in the KV cache can prevent the generation of new outlier tokens. I actually do not really understand whether there is a skipping mechanism in the autoregressive of generation or the authors suggest this would make the KV cache more quantization friendly. I would doubt the effectiveness of the method if they mean the later.
2. The performance without fine-tuning is actually not super strong, especially on the 70B models, it is actually maybe better if the author can change Table3 to add a column to indicate whether these other methods are fine-tuned or not so that the readers can understand the results better.
3. I doubt the run-time numbers in Table 5 continues to show advantages when the models are scaled to 70B. When models are memory-bound, whether it is a dynmaic/static quantization does not matter too much since most of the time are spent on loading the weights from HBM so that the arithmetic units on GPUs are under-utilized anyway. Do authros have results with 70B models, and do they still observe a speedup? If not, it is better to make sure these limitaitons are clearly addressed in the paper.

**Questions:**

1. Can you provide clarification on what do you really mean by "isolate outliers". Especially how would prefixing them in the KV cache help? In my understanding, the whole KV cache would have to join the decode stage computation, so you inevitably would have these outlier values even if you "prefix" them. Also in the autoregressive generation, naturally you will generate these tokens that are outliers too. I might have missed something obvious here, but I would like to have an explantion on this.
2. Can you indicate whether compared methods involve fine-tuning?
3. Can you show actual run time with large scale models? If no, what is the limitation of the proposed method?

---

> ### Author Response · Authors · 2024-11-14
> **Response to Reviewer wZKZ**
>
> Thank you for your comment. To address your concern, we reexplain how prefixquant works, and add the speedup results of 70B models.
>
>
> **Weakness 1 (Question 1):** It is not clear how the proposed method works. Specifically, I don't understand why prefixing certain outlier tokens in the KV cache prevents the generation of new outlier tokens. Is there a skipping mechanism in the autoregressive generation, or is the method intended to make the KV cache more quantization-friendly? I doubt the effectiveness if it's the latter.
>
> **Answer 1:** There seems to be a misunderstanding regarding our paper. In autoregressive inference, prefixing tokens in the input sequence or in the KV cache yields the same results. The advantage of prefixing in the KV cache is that it avoids corresponding computation on linear layers. Our method ensures outliers appear in initial tokens, which we then store in the KV cache to prevent their computation in down_proj layers. As mentioned in line 336, outliers only occupy a few tokens (1-4) per sequence. The positions of these outliers vary with the input sequence (see Figure 4b). By prefixing high-frequency outlier tokens, we confine them to prefixed positions (see Figure 4c). Thus, placing these tokens at the start of the KV cache achieves the desired outcome.
>
>
>
> **Weakness 2 (Question 2):** The performance without fine-tuning is not strong, especially on the 70B models. It would be helpful to add a column in Table 3 indicating whether other methods are fine-tuned, for better clarity.
>
> **Answer 2:** While the PrefixQuant model without fine-tuning is not exceptionally strong, it excels in efficient static quantization. Other methods suffer significant performance degradation under the same settings. For instance, W4A4KV4 Llama-3-8B with QuaRot in static quantization results in perplexity exceeding 100. In Table 3, Atom and DuQuant do not use fine-tuning, whereas QuaRot relies on additional GPTQ, and SpinQuant uses both fine-tuning and GPTQ. We will clarify this in the final version.
>
>
>
> **Weakness 3 (Question 3):** Do the authors have results with 70B models, and do they still observe a speedup? If not, these limitations should be clearly addressed.
>
> **Answer 3:** Table 9 in our paper presents the speedup of linear layers with 70B model shapes (8192x8192 and 8192x28672). Static quantization is faster than dynamic quantization. For end-to-end prefilling speedup, since the 4-bit Llama-3-70B can be loaded on an RTX 3090 GPU, we confirmed effectiveness on A100 GPUs only. As shown below, PrefixQuant achieves a 1.2x speedup over QuaRot.
>
> | Method| Speed|
> | ------------------ | -------------- |
> | FP16| OOM|
> | QuaRot (W4A4) | 1183 ms|
> | PrefixQuant (W4A4) | 993 ms (1.20x) |
>
> We sincerely appreciate your time and effort in reviewing our paper. Please let us know if you have further inquiries.

---

> ### Public Comment · ~Jeffrey_L_McKinstry1 · 2024-11-15
> **Still don't understand how this helps during inference, say on low precision hardware.**
>
> Very interesting result, but I don't understand how it can help just to insert outliers into the cache. Does this assume that your 4-bit KV cache has a separate scale factor for the prefixed tokens than for the rest of the entries in the cache?  Or are your outlier, prefixed cache entries somehow left in fp16 format?  In the latter case, this is not hardware friendly.  I would very much like to understand how these prepended tokens in the cache can have a dramatic affect on the subsequent responses without studying the code, as this is the key insight to your work, and thus the key idea to clarify to your readers.  Perhaps a bit of pseudocode as to how to implement the cache would help.
>
> I should say that my mental model of KV Cache is that all it does is store, in order, all the tokens generated so far so that it doesn't have to recompute them all when it generates the next token.  In this case, having precomputed certain tokens at the beginning doesn't help when you encounter them again in the middle of the prompt or response, such as '\n' or <space> tokens which may appear mutiple times within the sequence.

---

> > ### Author Response · Authors · 2024-11-21
> > **Response to the public comment from Jeffrey L McKinstry**
> >
> > Thanks for your interest.
> >
> > The main insight of our paper is that outlier tokens only occupy a few positions (1-4) per sequence. For instance, multiple '\n' tokens might be in the input, but only the first '\n' token becomes an outlier during inference, while the others remain at normal magnitudes.
> >
> > This means there are no outlier tokens in the input activation of linear layers through our PrefixQuant because they are stored in the KV cache and don’t need to be recalculated. For the KV cache, only the prefixed portion are outliers and  set to 16-bit, while subsequent kv cache are quantized to 4-bit. During inference, all quantized KV cache are de-quantized to 16-bit, allowing us to concatenate the prefixed and de-quantized KV cache efficiently.

---

> ### Comment · Reviewer_wZKZ · 2024-11-15
>
> >  Our method ensures outliers appear in initial tokens, which we then store in the KV cache to prevent their computation in down_proj layers. As mentioned in line 336, outliers only occupy a few tokens (1-4) per sequence. The positions of these outliers vary with the input sequence (see Figure 4b). By prefixing high-frequency outlier tokens, we confine them to prefixed positions (see Figure 4c). Thus, placing these tokens at the start of the KV cache achieves the desired outcome.
>
> I think I understand this part, my confusion comes from the fact that in the autoregressive generation phase, my model may start to generate new outlier tokens? For example, if I generate a new "\n" in my decoding phase, how do you add this to your KV cache?
>
> > Table 9 in our paper presents the speedup of linear layers with 70B model shapes (8192x8192 and 8192x28672)
>
> I don't think single layer profiling helps, do you have whole model perf results.

---

> > ### Author Response · Authors · 2024-11-21
> > **Response to Reviewer wZKZ**
> >
> > >  For example, if I generate a new "\n" in my decoding phase, how do you add this to your KV cache?
> >
> > Sorry for the confusion. I should reclaim that outliers only occupy a few positions (1-4) per sequence. For example, the input sequence may have 10 "\n" tokens. However,  only the first "\n" tokens would exist outliers in the intermediate activation. This why prefixed tokens in KV cache can work.
> >
> > > I don't think single-layer profiling helps, do you have whole model perf results?
> >
> > We test the 70B model prefilling speed with 2048 input length on A100-80GB. The results show that our method can achieve 2.1 x speedup with half the GPU number.
> >
> >
> >
> > | Method             | GPU number    | Latency （ms） |
> > | ------------------ | ------------- | -------------- |
> > | FP16               | 2 x A100 80GB | 1678 ms        |
> > | PrefixQuant (W4A4) | 1 x A100 80GB | 789 ms (2.1 x) |

---

### Official Review · Reviewer_Xnqz · 2024-11-02

**Soundness:** 3
**Presentation:** 3
**Contribution:** 1
**Rating:** 3
**Confidence:** 4

**Summary:**

This paper proposes a static activation quantization algorithm for large language models. The idea is to add prefix to the target LLM, which are selected in a way that it mitigates the outliers in other tokens so that the activations become more quantizable. The prefix are selected to be the top-k high-frequency outlier tokens. The method also applies Hadamard rotation and blockwise fine-tuning to further boost the performance. Experimental results suggest that the proposed method outperforms other dynamic quantization methods.

**Strengths:**

- I like the fact that the paper reports wall-clock inference speed on various devices (rtx 3090 and a100). This is missing from many quantization works, due to the difficulty of implementing kernels, but is nevertheless much needed.

- The presentation is clear and the visualizations are well-prepared.

- The generative quality of the method has been carefully measured, with many ablation studies.

**Weaknesses:**

- The biggest concern is the conceptual and technical novelty of the proposed method. As the authors mention in section 2, the idea of adding prefix tokens to mitigate the outliers has been already explored by two prior works: QFeP (Yang et al., arXived May 2024), and CushionCache (Son et al., EMNLP 2024). In particular, the central claim of this paper, i.e., such prefix makes the static quantization useful, has already been argued by CushionCache. If I understood correctly, it seems like the authors are claiming that there are two differences to these works. (1) PrefixQuant requires less computation than predecessors for optimizing the prefix, and (2) PrefixQuant outperforms these methods. The advantage (1) does not seem to be very critical practically (as these are one-time cost), and does not originate from a particularly technically novel component. The advantage (2) seems to come mainly from additionally considering Hadamard rotation, grid search, and block-wise fine-tuning, which are not original contributions of this paper. In fact, CushionCache already demonstrates that their method can be combined with Hadamard rotation to further boost the performance.

- It seems like the paper is claiming that the prefix plays a complementary role to Hadamard rotation, by arguing that Hadamard rotations are for addressing "channel-wise outliers" and the prefix are for addressing "token-wise outliers." However, I find this point very unclear and misleading, because previous empirical observations suggest that for many LLaMA-like models the outliers are localized in terms of both channels and tokens (e.g., Sun et al., COLM 2024). Thus, removing channel-wise outliers should also resolve token-wise outliers, logically. I request for a more concrete justification.

- The authors could have included evaluations on more realistic tasks, such as GSM-8k or MMLU.

- Looking at table 18, the claim that static per-tensor quantization by PrefixQuant outperforms existing dynamic quantization methods does not seem to be 100% true. At W8A8-like quantization on overparameterized models, i.e., with only very small degradation in performance, I still observe that QuaRot consistently outperforms PrefixQuant w/o FT. It seems likely that QuaRot+FT may also outperform PrefixQuant+FT.

**Questions:**

- Regarding the first weakness (above), I recommend the authors to compare the quality of their proposed prefix optimization method head-to-head with CushionCache and QFeP, by removing the grid search, Hadamard rotation, and block-wise fine-tuning.

---

> ### Author Response · Authors · 2024-11-14
> **Response to Reviewer Xnqz**
>
> Thank you for your comment.We reclaim the novelty of our paper, and explain why we need to remove token-wise outliers.
>
> **Weakness 1.1:** The main concern is the novelty of the proposed method. The idea of adding prefix tokens to mitigate outliers has been explored by previous works: QFeP (Yang et al., arXived May 2024) and CushionCache (Son et al., EMNLP 2024).
>
> **Answer 1.1:** Prefixed outlier tokens in KV cache have been explored in works like Massive Attention (Sun et al., COLM 2024) and Attention Sink (Xiao et al., ICLR 2024), alongside QFeP and CushionCache. However, PrefixQuant is the first to provide a system analysis of these outliers. As detailed in Section 4, we address upper outlier tokens in inputs and lower outliers in Q/K/V, which were not covered by previous works. Our method identifies outlier tokens within 10 seconds, compared to ~10 hours for CushionCache on Llama-3-8B. Additionally, PrefixQuant is the first to facilitate static quantization in both KV cache and activation.
>
>
> **Weakness 1.2:** The advantage of less computation time seems not very critical practically (as these are one-time costs) and does not stem from a technically novel component.
>
> **Answer 1.2:** We respectfully disagree. Compression time is crucial when deciding whether to adopt a compression technique. This is why methods like GPTQ and AWQ are popular—they complete compression quickly. Although finding the prefixed token is a one-time cost per model, the numerous models make this efficiency significant.
>
>
> **Weakness 1.3:** The advantage seems to come mainly from using Hadamard rotation, grid search, and block-wise fine-tuning, which are not original contributions. I recommend comparing the prefix optimization method with CushionCache and QFeP, excluding additional components.
>
> **Answer 1.3:** In head-to-head comparisons of prefixed tokens, all three papers—PrefixQuant, CushionCache, and QFeP—resolve token-wise outliers effectively, as shown in there activation distribution visualizations. However, PrefixQuant provides more comprehensive system analysis, covering both the KV cache and input of linear layers, unlike the others that focus solely on linear layers. Additionally, PrefixQuant identifies outliers 3600 times faster on Llama-3-8B (10s vs. ~10h) compared to CushionCache. We included Hadamard rotation in our comparisons because it has become a standard component used by several methods, including QuaRot, DuQuant, SpinQuant, and QoQ.
>
>
> **Weakness 2:** Removing channel-wise outliers should resolve token-wise outliers logically. I request a more concrete justification.
>
> **Answer 2:** Let me define outliers: `Channel-wise outliers` occur at fixed channel indexes, while `token-wise outliers` appear in specific tokens. Figure 1 in our paper illustrates this. As shown in Figure 1(a), outliers greater than 1,000 are present only in specific tokens, termed token-wise outliers. Figure 1(b) shows that after applying channel-wise alleviation methods like Rotation (QuaRot), outliers are redistributed across token channels. QuaRot reduces outliers to nearly 15 but still struggles with non-uniform distribution. Figure 1(c) demonstrates how our PrefixQuant isolates outlier tokens, reducing maximum values to nearly 0.07.
>
>
> **Weakness 3:** The authors could include evaluations on more realistic tasks, such as GSM-8k or MMLU.
>
> **Answer 3:** The table below shows our comparison results on the MMLU dataset, which is sensitive to quantization. QuaRot's performance collapses on MMLU, but PrefixQuant consistently outperforms previous dynamic quantization methods even without fine-tuning.
>
> | Model | Method| Quantization | Precision | MMLU Average Accuracy |
> | ---------- | ------------------ | ------------ | --------- | --------------------- |
> | LLama-3-8B | -| -| FP16 | 62.07 |
> | LLama-3-8B | QuaRot| Dynamic | w4A4KV4 | 34.25 |
> | LLama-3-8B | DuQuant| Dynamic | w4A4KV4 | 50.77 |
> | LLama-3-8B | SpinQuant| Dynamic | w4A4KV4 | 51.93 |
> | LLama-3-8B | PrefixQuant w/o FT | **Static** | W4A4KV4 | **53.02**|
> | LLama-3-8B | PrefixQuant| **Static** | W4A4KV4 | **54.65**|
> | LLama-3-8B | QuaRot| Dynamic | w4A8KV4 | 38.37 |
> | LLama-3-8B | DuQuant| Dynamic | w4A8KV4 | 58.01 |
> | LLama-3-8B | SpinQuant| Dynamic | w4A8KV4 | 58.25 |
> | LLama-3-8B | PrefixQuant w/o FT | **Static** | w4A8KV4 | **58.27**|
> | LLama-3-8B | PrefixQuant| **Static** | w4A8KV4 | **59.20**|
>
>
> **Weakness 4:** The claim that static per-tensor quantization by PrefixQuant outperforms existing dynamic methods does not seem entirely true.
>
> **Answer 4:** We detail comparison results in lines 465-480. We primarily claim that PrefixQuant outperforms existing methods in W4A4KV4 and W4A8KV4 settings. In the W8A8KV8 setting, PrefixQuant achieves comparable performance with existing methods, with its main advantage being efficient per-tensor static quantization.
>
> We sincerely appreciate the time dedicated to reviewing our paper. If you have further inquiries, please let us know.

---

> > ### Comment · Reviewer_Xnqz · 2024-11-26
> >
> > Sorry for the late reply. Here are some follow-up questions on the reviewer's response.
> >
> > **Comparison with QFeP and CushionCache.** Could authors give more details about what "system analysis" is? I am trying to see if this could account for the novelty concern. Also, regarding being first to quantize both activation and KV cache, I wonder if KV cache are automatically quantized, once the activations are quantized. Are authors referring to using even less bits?
> >
> > Also, regarding the point 1.3, I was asking for the experimental comparison, to be clear.
> >
> > **On MMLU.** I highly appreciate this experiment. Thanks.

---

> > > ### Author Response · Authors · 2024-11-26
> > >
> > > Thank you for your feedback. We address your follow-up questions below:
> > >
> > > > **Comparison with QFeP and CushionCache.** Could authors give more details about what "system analysis" is?
> > >
> > > 1. We propose a method to determine the number of outlier tokens by examining the original activation distribution (Lines 320–323). As shown in Table 1, we identify 4 outlier tokens for LLaMA-2-70B, while only 1 outlier token is sufficient for both LLaMA-3-8B and Qwen-2-7B. In contrast, QFeP and CushionCache treat the number of outlier tokens as a hyperparameter, setting it to 3 for all models.
> > > 2. We reveal that outlier tokens occur only in certain special tokens (Figure 4(a)). Using this insight, we directly identify the prefixed outlier tokens by analyzing high-frequency occurrences. On the other hand, QFeP and CushionCache rely on grid searching the entire vocabulary of the LLM, which is significantly more time-consuming. For instance, our method identifies outlier tokens in approximately 10 seconds, compared to ~10 hours for CushionCache on LLaMA-3-8B.
> > > 3. We are the first to uncover lower outliers in Q/K, whereas previous works have only focused on upper outliers in down_proj layers.
> > > 4. Finally, we emphasize that ours is the first work to demonstrate that static quantization can outperform dynamic quantization in LLMs.
> > >
> > >
> > >
> > > > Also, regarding being first to quantize both activation and KV cache, I wonder if KV cache are automatically quantized, once the activations are quantized.
> > >
> > > KV cache quantization and activation quantization are distinct processes. For activation quantization, we quantize the inputs to linear layers while keeping the outputs of these layers in full precision. In contrast, KV cache quantization involves quantizing the outputs of the k_proj and v_proj layers into low-bit representations.
> > >
> > >
> > >
> > > > Also, regarding the point 1.3, I was asking for the experimental comparison, to be clear.
> > >
> > > We have designed fair experimental settings to compare prefix optimization algorithms:
> > >
> > > 1. **Comparison with CushionCache.** CushionCache neither open-sourced their code nor recorded the prefixed tokens in their paper. To enable fair comparison, we deactivate rotation in our method and perform W8A8 per-tensor static quantization. We retain fine-tuning since CushionCache also performed fine-tuning (see Table 6 in their paper).  As shown in the table below, PrefixQuant achieves lower WikiText-2 perplexity compared to CushionCache. Moreover, PrefixQuant identifies the prefixed tokens 3,600× faster than CushionCache (12 seconds vs. 12 hours).
> > >
> > > | Model      | Method       | Wiki PPL | Prefix Time     |
> > > | ---------- | ------------ | -------- | --------------- |
> > > | Llama-2-7B | CushionCache | 5.98     | 2.68 hours      |
> > > | Llama-2-7B | PrefixQuant  | **5.73** | **11 seconds**  |
> > > | Llama-3-8B | CushionCache | 7.41     | 12.09 hours     |
> > > | Llama-3-8B | PrefixQuant  | **7.22** | **12  seconds** |
> > >
> > > 2. **Comparison with QFeP.** QFeP provides the prefixed tokens in their paper. To ensure a fair comparison, we directly replace the prefixed outlier tokens in our method with those provided by QFeP. Note that these experiments do not involve fine-tuning. Unfortunately, we cannot compare the time required to search for prefixed tokens in QFeP because this information is not provided in their paper or open-sourced code. As shown in the table below, PrefixQuant consistently outperforms QFeP in W4A4KV4 settings, using the same configuration apart from the prefixed tokens, and achieves this with significantly less prefixed token search time.
> > >
> > > | Model       | Method      | Prefixed Number | Prefixed tokens | Wiki PPL | Prefix Time |
> > > | ----------- | ----------- | --------------- | --------------- | -------- | ----------- |
> > > | Llama-2-7B  | QFeP        | 3               | [BOS] all .     | 8.11     | >> 11 s     |
> > > | Llama-2-7B  | PrefixQuant | 3               | .\n[BOS]        | **6.01** | 11 s        |
> > > | Llama-2-13B | QFeP        | 3               | [BOS] then ,    | 7.66     | >> 20 s     |
> > > | Llama-2-13B | PrefixQuant | 3               | the.[BOS]       | **5.50** | 20 s        |
> > > | Llama-2-70B | QFeP        | 3               | [BOS] I ’       | 4.53     | >> 60 s     |
> > > | Llama-2-70B | PrefixQuant | 4               | \n”[BOS]        | **4.41** | 60 s        |
> > > | Llama-3-8B  | QFeP        | 3               | [BOS] - nd      | 7.95     | >> 12 s     |
> > > | Llama-3-8B  | PrefixQuant | 1               | [BOS]           | **7.93** | 12 s        |
> > > | Llama-3-70B | QFeP        | 3               | [BOS] and ,     | 5.32     | >> 60 s     |
> > > | Llama-3-70B | PrefixQuant | 3               | , [BOS]         | **5.23** | 60 s        |

---

> > > ### Author Response · Authors · 2024-11-29
> > > **Looking forward to your reply !**
> > >
> > > We sincerely thank you for your insightful suggestions on this paper. In response, we have carefully addressed your points with detailed explanations. Should you have any additional questions, we would be happy to provide further clarifications or conduct additional experiments. We look forward to your feedback!

---

### Official Review · Reviewer_xJQ6 · 2024-11-04

**Soundness:** 1
**Presentation:** 2
**Contribution:** 1
**Rating:** 3
**Confidence:** 4

**Summary:**

The authors proposed PrefixQuant, which allows for efficient per-tensor static quantization to outperform expensive per-token dynamic quantization. They showed that PrefixQuant with per-tensor static quantization can outperform previous per-token dynamic quantization methods like QuaRot.

**Strengths:**

- The authors showed the possibility that per-tensor static quantization can outperform per-token dynamic quantization.

- They measured the real time-to-first-token (pre-filling) speed-up.

**Weaknesses:**

(1) The authors merely showed the effectiveness of PrefixQuant with per-tensor static quantization only $\textbf{when the context length is 2048}$ (Table 2, 3, 4, 5, and 6). Since 2048 context length is relatively short, per-tensor static quantization might work. However, when the context length is 8192, for example, the activation size would be 8192 (context length) $\times$ 4096 (model hidden size) = 33554432. Then, even if using 8-bit per-tensor static activation quantization, 33554432 / 256 (8-bit) = 131072 numbers have to be represented in only a single integer on average, which would naturally incur more severe quantization error than when the context length is 2048. In other words, in the case of per-tensor static activation quantization, as the context length goes longer, the larger numbers have to be represented in only a single integer on average, thus causing per-tensor static quantization to perform worse.

However, in the case of per-token dynamic quantization, no matter how long the context length is, just 4096 (model hidden size) / 256 (8-bit) = 16 numbers have to be represented in only a single integer on average. Considering that many long-context LLMs are sought-after these days, it is necessary to compare PrefixQuant with per-tensor static quantization with previous per-token dynamic quantization methods like QuaRot when the context length is 8192 or longer. Without the comparison in a long-context setting, it is not convincing that PrefixQuant is the first to enable efficient per-tensor static quantization to outperform expensive per-token dynamic quantization (mentioned in Abstract).

(2) The paper focuses on perplexity and common sense reasoning tasks as the performance measure. More experiments are required to assess the effectiveness of the proposed method on broader challenging subjects like MMLU.

**Questions:**

It would be better if the authors measured the real time-to-first-token (pre-filling) speed-up with longer context length (e.g., 8192) than 2048.

---

> ### Author Response · Authors · 2024-11-14
> **Response to reviewer xJQ6**
>
> Thank you for your comment. To address your concern, we have added performance comparisons in a long-context setting (8192 context length). This demonstrates that per-tensor quantization of PrefixQuant can outperform previous per-token dynamic quantization methods. We also present the MMLU results and speedup with an 8192 long context.
>
> **Weakness 1:** Lack of comparison in a long-context setting
>
> **Answer 1:** The following table shows that PrefixQuant, without fine-tuning, outperforms previous per-token dynamic quantization methods in both 4-bit and 8-bit activations at a context length of 8192 (the maximum length of the original Llama-3-8B). These results demonstrate that PrefixQuant is the first to enable efficient per-tensor static quantization, outperforming the costly per-token dynamic quantization.
>
> | Model      | Method             | Sequence Length | Quantization | Precision | WikiText2 PPL. |
> | ---------- | ------------------ | --------------- | ------------ | --------- | -------------- |
> | LLama-3-8B | -                  | 8192            | -            | FP16      | 5.54           |
> | LLama-3-8B | QuaRot             | 8192            | Dynamic      | W4A8KV4   | 6.79           |
> | LLama-3-8B | PrefixQuant w/o FT | 8192            | **Static**   | W4A8KV4   | **6.21**       |
> | LLama-3-8B | PrefixQuant        | 8192            | **Static**   | W4A8KV4   | **6.04**       |
> | LLama-3-8B | QuaRot             | 8192            | Dynamic      | w4A4KV4   | 8.41           |
> | LLama-3-8B | DuQuant            | 8192            | Dynamic      | w4A4KV4   | 7.27           |
> | LLama-3-8B | PrefixQuant w/o FT | 8192            | **Static**   | w4A4KV4   | **7.13**       |
> | LLama-3-8B | PrefixQuant        | 8192            | **Static**   | w4A4KV4   | **6.82**       |
>
> **Weakness 2:** More experiments are needed to assess the effectiveness on challenging subjects like MMLU.
>
> **Answer 2:** The following table illustrates the comparison results on MMLU datasets, which are more sensitive to quantization. It shows that the performance of QuaRot collapses on the MMLU dataset. However, PrefixQuant consistently outperforms previous dynamic quantization methods, even without fine-tuning.
>
> | Model      | Method             | Quantization | Precision | MMLU Average Accuracy |
> | ---------- | ------------------ | ------------ | --------- | --------------------- |
> | LLama-3-8B | -                  | -            | FP16      | 62.07                 |
> | LLama-3-8B | QuaRot             | Dynamic      | w4A4KV4   | 34.25                 |
> | LLama-3-8B | DuQuant            | Dynamic      | w4A4KV4   | 50.77                 |
> | LLama-3-8B | SpinQuant          | Dynamic      | w4A4KV4   | 51.93                 |
> | LLama-3-8B | PrefixQuant w/o FT | **Static**   | W4A4KV4   | **53.02**             |
> | LLama-3-8B | PrefixQuant        | **Static**   | W4A4KV4   | **54.65**             |
> | LLama-3-8B | QuaRot             | Dynamic      | w4A8KV4   | 38.37                 |
> | LLama-3-8B | DuQuant            | Dynamic      | w4A8KV4   | 58.01                 |
> | LLama-3-8B | SpinQuant          | Dynamic      | w4A8KV4   | 58.25                 |
> | LLama-3-8B | PrefixQuant w/o FT | **Static**   | w4A8KV4   | **58.27**             |
> | LLama-3-8B | PrefixQuant        | **Static**   | w4A8KV4   | **59.20**             |
>
> **Question 3:** It would be better if the authors measured the real time-to-first-token (pre-filling) speed-up with longer context length (e.g., 8192) than 2048.
>
> **Answer 3:** We tested the W4A4 Llama-3-8B pre-filling speedup compared to FP16 with a batch size of 1 and a context length of 8192. As shown in the table below, PrefixQuant achieves a 1.83x speedup on the A100 and a 3.02x speedup on the RTX 3090.
>
> | GPUs      | W4A4 vs. FP16 Speedup Ratio |
> | --------- | --------------------------- |
> | RTX 3090  | 3.02x                       |
> | A100-80GB | 1.83x                       |
>
> We sincerely appreciate the time and effort you have dedicated to reviewing our paper. Should you have any further inquiries, please let us know.

---

> > ### Author Response · Authors · 2024-11-29
> > **Looking forward to your reply !**
> >
> > We sincerely thank you for your insightful suggestions on this paper. In response, we have carefully addressed your points with detailed explanations. Should you have any additional questions, we would be happy to provide further clarifications or conduct additional experiments. We look forward to your feedback!

---

> > > ### Comment · Reviewer_xJQ6 · 2024-11-30
> > >
> > > Thank you for the detailed response. The experimental results the author provided are interesting, but my major concerns have not been addressed yet due to the following reasons.
> > >
> > > (1) Table in Answer 1 does not include the experimental results of DuQuant and SpinQuant in W4A8KV4 and those of SpinQuant in W4A4KV4.
> > >
> > > (2) Furthermore, the authors do not conduct experiments of W8A8KV8 in Answer 1.
> > >
> > > (3) As I mentioned in the review, more longer the context length is, more severe accuracy degradation per-tensor static activation quantization incurs. However, there is no demonstration of why PrefixQuant can work better than per-token dynamic activation quantization in a long-context setting. As a consequence, in order to validate the efficacy of PrefixQuant in a long-context setting only with empirical results, I believe that PrefixQuant has to be compared to QuaRot, DuQuant, and SpinQuant for various quantization schemes (e.g., W8A8KV8, W4A8KV8, W4A8KV4) with the context length ranging from 8192 to 128K (if possible). 'Needle In A Haystack' might be a good choice for this.
> > >
> > > Accordingly, I keep my original score until now.

---

> > > > ### Author Response · Authors · 2024-12-01
> > > >
> > > > We thank you for your feedback.
> > > >
> > > >
> > > >
> > > > **Additional comparisons results**: As shown in the following table, we compare PrefixQuant with QuaRot, DuQuant, and SpinQuant under various quantization schemes with a context length of 8192. The results show that PrefixQuant outperforms previous methods in most cases, except for W8A8KV8 with SpinQuant.
> > > >
> > > > **Why static can outperform dynamic**: Dynamic quantization can adapt to the varying activation ranges of different tokens. However, it struggles to balance clipping and rounding errors during quantization, as it simply applies Min-Max quantization to each token. As shown in Figure 2(c), the token distributions across different tokens become more uniform after PrefixQuant. In such scenarios, balancing clipping and rounding errors becomes more critical than adapting to token-specific distributions for maintaining performance. This explains why PrefixQuant enables static quantization to outperform dynamic quantization.
> > > >
> > > > **Needle in a Haystack**: Regarding the experiments on "Needle in a Haystack," this is beyond the primary scope of a quantization algorithm paper. Nevertheless, we appreciate your suggestion and will consider conducting these experiments. However, due to time constraints, it may not be feasible to include them in the current version.
> > > >
> > > >
> > > >
> > > > | Model      | Method             | Seq. Length | Quant.     | Precision | WikiText2 PPL. |
> > > > | ---------- | ------------------ | ----------- | ---------- | --------- | -------------- |
> > > > | LLama-3-8B | -                  | 8192        | -          | FP16      | 5.54           |
> > > > | LLama-3-8B | QuaRot             | 8192        | Dynamic    | W8A8KV8   | 6.15           |
> > > > | LLama-3-8B | DuQuant            | 8192        | Dynamic    | W8A8KV8   | 5.60           |
> > > > | LLama-3-8B | SpinQuant          | 8192        | Dynamic    | W8A8KV8   | **5.56**       |
> > > > | LLama-3-8B | PrefixQuant w/o FT | 8192        | **Static** | W8A8KV8   | **5.56**       |
> > > > | LLama-3-8B | QuaRot             | 8192        | Dynamic    | W4A8KV4   | 6.79           |
> > > > | LLama-3-8B | DuQuant            | 8192        | Dynamic    | W4A8KV4   | 6.19           |
> > > > | LLama-3-8B | SpinQuant          | 8192        | Dynamic    | W4A8KV4   | 6.15           |
> > > > | LLama-3-8B | PrefixQuant w/o FT | 8192        | **Static** | W4A8KV4   | **6.21**       |
> > > > | LLama-3-8B | PrefixQuant        | 8192        | **Static** | W4A8KV4   | **6.04**       |
> > > > | LLama-3-8B | QuaRot             | 8192        | Dynamic    | w4A4KV4   | 8.41           |
> > > > | LLama-3-8B | DuQuant            | 8192        | Dynamic    | w4A4KV4   | 7.27           |
> > > > | LLama-3-8B | SpinQuant          | 8192        | Dynamic    | w4A4KV4   | 7.23           |
> > > > | LLama-3-8B | PrefixQuant w/o FT | 8192        | **Static** | w4A4KV4   | **7.13**       |
> > > > | LLama-3-8B | PrefixQuant        | 8192        | **Static** | w4A4KV4   | **6.82**       |

---

> ### Author Response · Authors · 2024-11-26
> **Looking forward to feedback**
>
> Dear Reviewer xJQ6,
>
> We have added performance comparisons in a long-context setting (8192 context length) to address your primary concern. Additionally, we would like to emphasize that static quantization is particularly practical, especially for edge devices such as NPUs, which often do not support dynamic quantization. We look forward to your feedback and appreciate your time and consideration.

---

### Meta-Review · Area_Chair_5fzS · 2024-12-22

**Metareview:**

This paper introduces PrefixQuant, a static quantization method for LLMs leveraging prefixed outliers in the KV cache. While the paper provides efficiency improvements and novel contributions, the reviewers have identified some key weaknesses. Specifically, the conceptual novelty is limited, with significant overlap with prior work like CushionCache and QFeP. Critical comparisons and evaluations on long-context settings and diverse tasks are insufficient, reducing confidence in its claims. The technical explanation of core mechanisms, such as how prefixing prevents new outliers, lacks clarity. Despite the authors’ rebuttal, the paper does not currently meet the bar for significant advancement in quantization methods. The reviewers unanimously agree on the rejection.

**Additional Comments On Reviewer Discussion:**

The reviewers raised concerns regarding technical novelty, long-context evals, and the lack of comparison with baselines such as CushionCache. While the authors provided additional experiments and provided further clarification, these did not fully address the reviewer concerns. For instance, long-context results still lack broader quantization settings. Perhaps more importantly, technical novelty remains a critical issue due to overlap with existing methods.

---

### Decision · Program_Chairs · 2025-01-22

Reject